# Global landscape of replicative DNA polymerase usage in the human genome

Eri Koyanagi[1,10], Yoko Kakimoto[1,10], Tamiko Minamisawa[2,10], Fumiya Yoshifuji[3,10], Toyoaki Natsume [4,5,9], Atsushi Higashitani [3], Tomoo Ogi [6], Antony M. Carr [7], Masato T. Kanemaki [4,5,8] & Yasukazu Daigaku [1,2]

The division of labour among DNA polymerase underlies the accuracy and efficiency of replication. However, the roles of replicative polymerases have not been directly established in human cells. We developed polymerase usage sequencing (Pu-seq) in HCT116 cells and mapped Polε and Polα usage genome wide. The polymerase usage profiles show Polε synthesises the leading strand and Polα contributes mainly to lagging strand synthesis. Combining the Polε and Polα profiles, we accurately predict the genome-wide pattern of fork directionality plus zones of replication initiation and termination. We confirm that transcriptional activity contributes to the pattern of initiation and termination and, by separately analysing the effect of transcription on co-directional and converging forks, demonstrate that coupled DNA synthesis of leading and lagging strands is compromised by transcription in both co-directional and convergent forks. Polymerase uncoupling is particularly evident in the vicinity of large genes, including the two most unstable common fragile sites, FRA3B and FRA3D, thus linking transcription-induced polymerase uncoupling to chromosomal instability. Together, our result demonstrated that Pu-seq in human cells provides a powerful and straightforward methodology to explore DNA polymerase usage and replication fork dynamics.

Accurate DNA replication underlies stable genetic inheritance and is essential in all eukaryotic organisms. In humans, the loss of replication fidelity is responsible for genetic changes that cause both inherited syndromes and somatic diseases, including cancer. There are 16 different DNA polymerases in eukaryotes and the fidelity and efficiency of their synthetic activities are distinct[1]. The division of labour among these polymerases is, therefore, a primary factor in determining the accuracy of genome duplication. In both the budding and fission yeasts, three DNA polymerases: Polδ, Polε and Polα, have been demonstrated to be required for genome replication and are thus termed replicative polymerases. To start all canonical replication events, primase initiates a short RNA primer that is subsequently extended for 10–20 nucleotides by Polα. On the leading strand the bulk of DNA synthesis is subsequently completed by Polε. On the lagging strand, where synthesis is by necessity discontinuous, Polδ takes over from Polα to extend the synthesis up to 100 to 200 bp, generating the Okazaki fragment.

[1]Frontier Research Institute for Interdisciplinary Sciences, Tohoku University, Sendai, Japan. [2]Cancer Genome Dynamics project, Cancer Institute, Japanese Foundation for Cancer Research, Tokyo, Japan. [3]Graduate School of Life Sciences, Tohoku University, Sendai, Japan. [4]National Institute of Genetics, Research Organization of Information and Systems (ROIS), Mishima, Japan. [5]Department of Genetics, The Graduate University for Advanced Studies (SOKENDAI), Mishima, Japan. [6]Research Institute of Environmental Medicine, Nagoya University, Nagoya, Japan. [7]Genome Damage and Stability Centre, School of Life Sciences, University of Sussex, Falmer BN1 9RQ, UK. [8]Department of Biological Sciences, Graduate School of Science, The University of Tokyo, Tokyo, Japan. [9]Present address: Research Center for Genome & Medical Sciences, Tokyo Metropolitan Institute of Medical Science, Tokyo, Japan. [10]These authors contributed equally: Eri Koyanagi, Yoko Kakimoto, Tamiko Minamisawa, Fumiya Yoshifuji. e-mail: yasukazu.daigaku@jfcr.or.jp

The roles of the replicative polymerases were established in budding yeast from the mutational bias caused by altered (mutagenic) replicative polymerases in the vicinity of an efficient replication origin, where replication directionality could be predicted[2,3]. Using a similar mutational bias approach in fission yeast, the role of Polδ in lagging strand synthesis was shown to be conserved. Using a mutated Polε that is prone to introducing ribonucleotides (rNMPs) into DNA it was also shown, using strand-specific alkali sensitivity, that the role of Polε in synthesising the leading strand was similarly conserved[4]. To expand the analysis of polymerase usage genome-wide, the locations of the increased levels of rNMPs incorporated by individual mutated DNA polymerases (Polα, Polδ or Polε) were identified by whole genome sequencing[5–8]. These data provided direct evidence that the bulk of leading strand synthesis is performed by Polε, while that of the lagging strand was the responsibility of Polα and Polδ. The signature of rNMP incorporation by wild-type polymerases also confirmed roles of these polymerases[9]. Consistent with these in vivo reports, the roles of budding yeast replicative polymerases have similarly been demonstrated by in vitro studies that reconstituted the replisome with purified factors[10,11].

In addition to confirming the division of labour among replicative DNA polymerase, the genome-wide data of replicative polymerase usage also provided highly detailed and discriminatory information about replication fork dynamics. For example, genomic sites with an increased probability of either replication initiation or termination associate with reciprocal changes in leading and lagging strand polymerase usage. By calculating relative changes (differential derivatives) of the profiles of the individual replicative polymerases, the population percentage of replication initiation and termination events were globally measured. This approach identified replication initiation sites, plus their probability of initiation (efficiency), at unpreceded resolution in both budding and fission yeast. It also provided an estimation of the probability of termination across the genome[5,12]. The accuracy of the methodology was exemplified by the fact that initiation sites identified in budding yeast correspond with the known sequence-specificity of replication origins. In fission yeast, the initiation sites correlated closely with AT richness[13]. Indeed, Monte Carlo simulation of replication fork dynamics based solely on the predicted distribution of the origin recognition complex, which was calculated from the genomic AT content, produced a profile of fork dynamics that was strikingly similar to the experimental data[14].

The profiles of leading and lagging strand DNA polymerase usage can also be directly converted into replication fork directionality (RFD), which represents the proportion of leftward or rightward moving forks at each genomic locus. Again, the accuracy of these data has been verified by multiple studies. For example, mathematical analysis of the RFD data derived from polymerase usage has been used to predict replication timing (RT) across entire chromosomes. The resulting data is superimposable on experimental RT data derived from measured DNA copy number[5]. Given the precision, quantitative accuracy and the concordance with previous measurement of replication fork dynamics, the identification of polymerase usage provides a rational approach to explore genome replication globally in other eukaryotic organisms.

In contrast to lower eukaryotes, the division of labour among replicative polymerases in metazoan cells remains to be addressed. Although human Polα and Polδ have been shown to be required for the synthesis of both leading and lagging strands in reconstituted SV40 replication systems[15], the usage of replicative polymerases during genomic DNA replication has not been directly characterised. To elucidate the division of labour among replicative polymerases in human cells, and to analyse the profile of replication forks at high resolution, we set out to track the usage of leading and lagging strand DNA polymerases across the human genome. In order to track synthesis by replicative DNA polymerases, we followed the equivalent logic of polymerase usage sequencing (Pu-seq, also known as HydEn-seq) in

the yeasts and exploited alleles of replicative DNA polymerases that incorporate an excess of rNMP into DNA during synthesis[5–7] (Fig. 1a).

Using the near-diploid colon cancer cell line HCT116 we successfully produced genome-wide profiles of Polε and Polα usage that, respectively, reflected leading and lagging strand synthesis. By analysing these profiles, we confirm that transcriptional activity influences replication initiation, demonstrate that fork directionality impacts termination close to transcription start sites and show that transcription can perturb the coupling of leading and lagging strand polymerases. Finally, we also show that, at several common fragile sites (CFS) expressed in HCT116 cells, a high level of uncoupled polymerase usage is apparent due to the local inhibition of leading strand DNA synthesis.

## Results

### Construction of ribonucleotide-incorporating DNA polymerase mutant lines

Ribonucleotides are normally incorporated by the replicative polymerases approximately 1:500–5000 incorporation events depending on the polymerase. The mutated alleles of replicative polymerases used in yeast to map polymerase usage increase this rate of incorporation by more than 10-fold[16]. Nonetheless, such rNMPs are rapidly removed from duplex DNA by ribonucleotide excision repair (RER), which is initiated by the RNase H2 enzyme. Unlike in yeast, RNase H2 is essential for growth of mammalian cells in the presence of p53[17]. Thus, we developed a HCT116 cell line where we could induce acute degradation of the largest RNase H2 subunit, RNASEH2A, by the addition of auxin. In this auxin-inducible degron (AID) system[18] the target protein, RNASEH2A, is tagged with the minimal auxin-inducible degron tag (mAID) and an auxin receptor protein from rice (OsTIR1) is expressed. Treatment of the cells with the plant hormone 3-indole-acetic acid (IAA) promotes an interaction between OsTIR1 (an Cullin E3 ubiquitin ligase component) and the mAID-tagged RNASEH2A. Recruiting mAID tagged RNASEH2A to the Cullin E3 ligase by the addition of IAA induced ubiquitylation-dependent degradation of the targeted protein (Fig. 1b, left).

To generate mutant alleles of the POLD1, POLE1 and POLA1 genes (encoding the catalytic subunits of Polδ, Polε and Polα respectively) that are predicted to promote increased rNMP incorporation during synthesis, we aligned the highly conserved amino acid sequence of their catalytic sites with the corresponding yeast polymerases. This identified POLD1-L606G or L606M, POLE1-M630F and POLA1-Y865F as equivalent mutations to those exploited in budding and fission yeasts (Fig. 1c). Using CRISPR-Cas9-mediated homology-directed repair, we attempted to generate bi-allelic mutations in the RNASEH2A-degron HCT116 cells. As a result, POLE1-M630F and POLA1-Y865F mutations were independently introduced into both alleles of the corresponding gene. The POLD1-L606G or L606M mutation was, however, only introduced into the one allele, even after repetitive trials, suggesting that the biallelic mutation encoding these amino acid substitutions caused lethality.

Using the biallelic mutant cell lines for Polε and Polα, in addition to relevant control cell lines, we examined whether RNASEH2A degradation causes increased levels of rNMP in the DNA. Genomic DNA was extracted, treated with alkaline (which preferentially hydrolyses the phosphate-backbone 3′ of the incorporated rNMPs) and the extent of fragmentation was examined by running the denatured samples on agarose gels[5,6]. For the POLE1-M630F and POLA1-Y865F cell lines, increased levels of small DNA fragments were observed upon RNASEH2A degradation when compared control cell lines (Fig. 1d, e, Supplementary Fig. 1a). This indicates that rNMPs are incorporated at appreciably higher levels by the mutated Polε and Polα. However, rNMP incorporation in the POLA1-Y865F cell line was limited in comparison to the POLE1-M630F cell line. We therefore adapted the POLA1-Y865F cell line to the recently developed AID2 system that exploits the highly specific binding of a mutated version of OsTIR1 (F74G) with an auxin analogue, 5-Ph-IAA[19]. As expected, more efficient degradation of

RNASEH2A was observed (Fig. 1b right) and considerably increased levels of incorporated rNMP were evident in the POLA1-Y865F cell line when compared to controls (Fig. 1f).

We next tested whether RNASEH2A degradation affected cell growth and replication dynamics over 48 h. We did not observe a reduced rate of cell growth or a decrease in DNA synthesis in response to RNASEH2A degradation in either polymerase mutants or wild type polymerase cells (Supplementary Fig. 1b, c). In addition, we confirmed that the structures recognised by the S9.6 antibody proposed to represent DNA/RNA hybrids (i.e. are RNase H sensitive in vitro) were not increased following RNASEH2A degradation (Supplementary Fig. 1d). We also demonstrated that the DNA damage checkpoint was not activated by either the presence of mutant polymerases, the degradation of RNASEH2A or a combination of both (Supplementary Fig. 1e). These results show that, 48 h following treatment (i.e. when

cells are sampled), replication dynamics are not significantly influenced.

## Mapping polymerase usage across the genome

To map usage of Polε and Polα, DNA was prepared from the relevant cell lines (POLE1-M630F, POLA1-Y865F and POL+ all in an RNASEH2A-mAID background) 48 h after IAA/5-Ph-IAA addition and small alkaline-cleaved single stranded-DNA (ssDNA) fragments (<2 kb) were collected and used to produce libraries for Illumina sequencing. Approximately 200 million paired-end reads were obtained per experiment for each cell line and the positions of 5′ ends were mapped to either the Watson or Crick strands. The 5′ ends represent the rNMP positions, which were scored in 1-kb bins across the genome (Supplementary Fig. 2). The relative ratio of reads for Polε and Polα mutants with RNASEH2A degradation, when compared to those of control lines (POL+ with

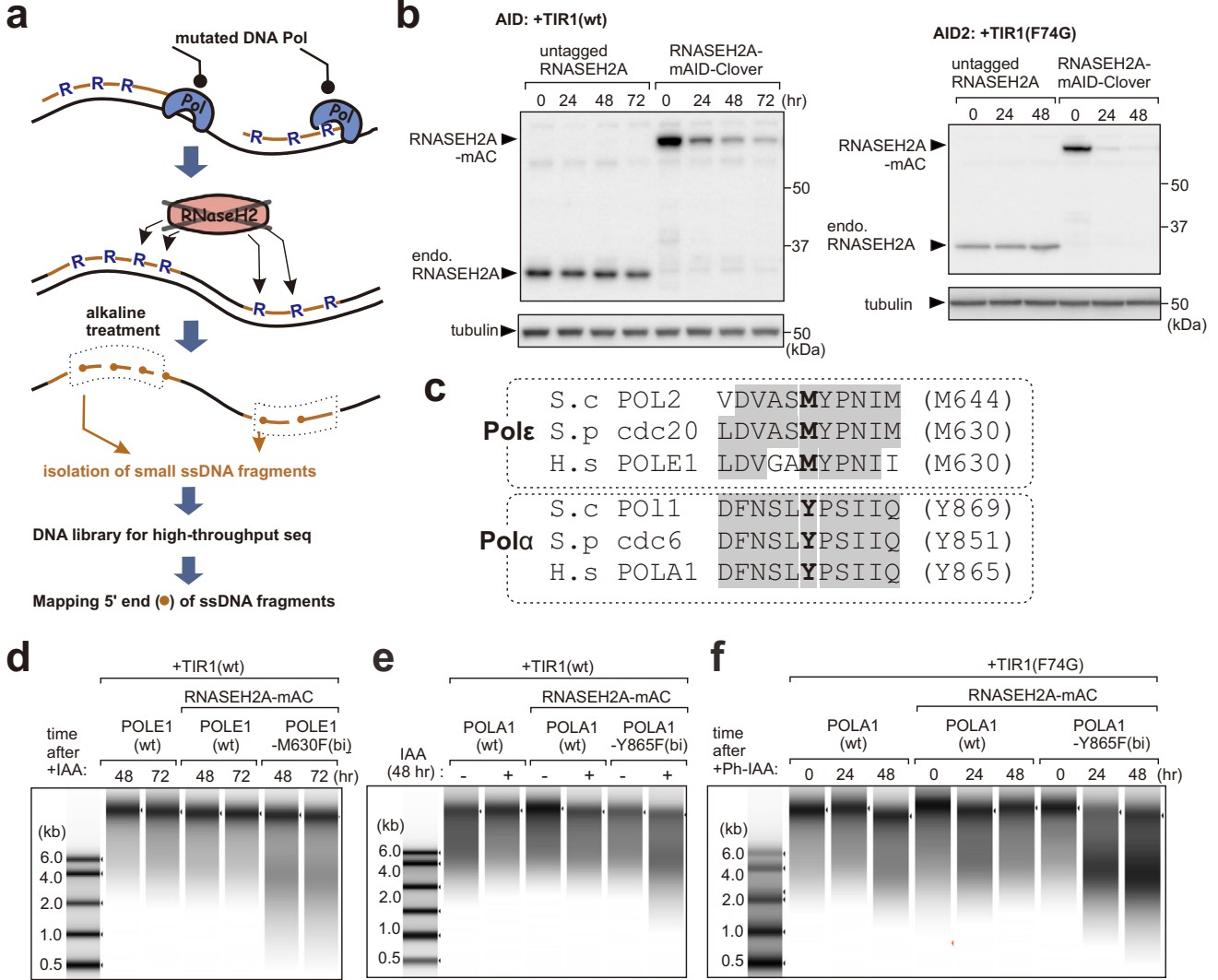

**Fig. 1 | Ribonucleotide incorporation into DNA in POLE1-M630F and POLA1-Y865F cells. a** Schematic representation of Pu-seq. Top: ribonucleotides (R) are incorporated by the mutated DNA polymerase. Ochre lines indicate the part of DNA synthesis by the mutated polymerase. Middle: in the absence of RNase H2-dependent RER, rNMPs remain in the DNA. Bottom: the sugar backbone of DNA strand is cleaved at sites of rNMP incorporation by alkali. Small ssDNA fragments (dashed box) are collected and subjected to library preparation and sequencing to identify the 5′ end of ssDNA fragment (circled end of ochre lines) as the location of rNMP. **b** Left: auxin-induced degradation of RNASEH2A following addition of indole-3-acetic acid (IAA) to cells expressing wild type *O. sativa* TIR1 (AID system).

Right: 5-Ph-IAA-induced degradation of RNASH2A in cells expressing *O. sativa* TIR1(F74G) (AID2 system). We repeated this experiment three times and obtained similar results. **c** Conservation of targeted amino acid residues of DNA Polε and Polα to induce rNMP incorporation. **d**–**f** Determination of rNMP incorporation into genomic DNA. Extracted genomic DNA from the indicated cell lines after the initiation of RNASEH2A degradation was treated with alkali to cleave at incorporated rNMP and analysed by electrophoresis. Source data of (**b**–**f**) are provided as a Source Data file. We repeated experiments in (**b**–**f**) three times and obtained similar results (see replicated results of (**d**) and (**f**) in Supplementary Fig. 1a).

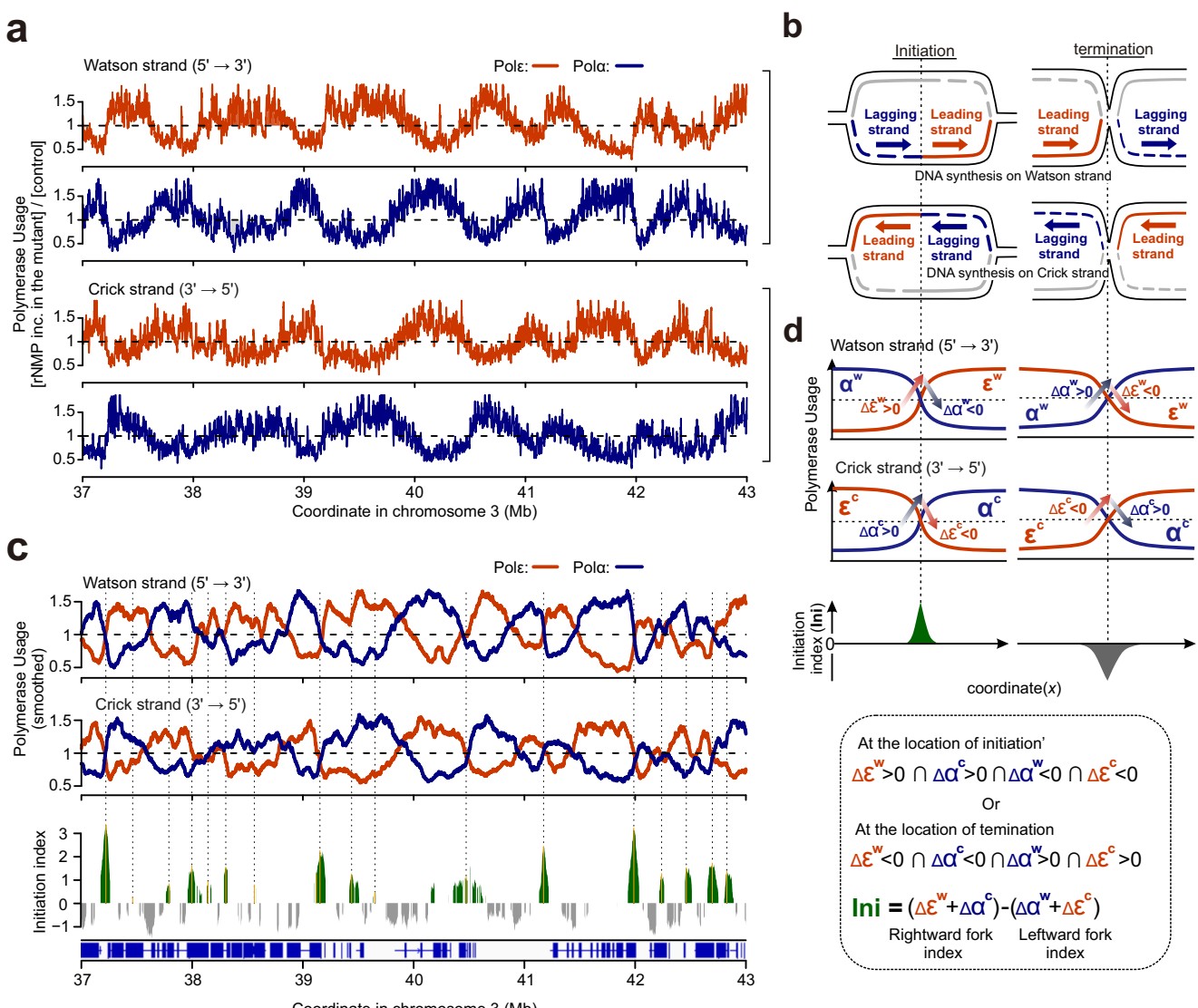

**Fig. 2 | Polymerase usage and replication initiation across the human genome.**
**a** Profiles of the relative reads for Polε and Polα mutants on the Watson and Crick strand for each 1 kb bin. Orange: Polε (POLE1-M630F). Blue: Polα (POLA1-Y865F). A representative region of chromosome 3 is shown. Data were smoothed with a moving average ($m = 3$; see "Materials and Methods"). **b** Schematic representation of predicted Polε and Polα profiles at a site of replication initiation and termination. Orange: leading strand. Blue: lagging strand. **c** Top: smoothed data from panel a (moving average, $m = 30$) provides a map of polymerase usage. Bottom: plot of the calculated initiation index. Positive values (green) represent increased initiation activity. Negative values (grey) represent increased converging fork termination. **d** Definition of the initiation index (see materials and methods for further details). 'Δ' indicates the differential between neighbouring bins, e.g. at location $x$, $\Delta\varepsilon(x)$ is defined as $\varepsilon(x+1)\cdot\varepsilon(x)$.

RNASEH2A degradation), provides scores representative of relative usage of these polymerases (Fig. 2a). Along the chromosome coordinates, a reciprocal relationship was evident between the profiles of Polε and Polα on the same strand. Similarly, a reciprocal relationship was evident between the profiles of the same polymerase when comparing the Watson with the Crick strand. These patterns of polymerase profiles are evident across the genome, consistent with the primary roles of Polε in leading strand and Polα in lagging strand synthesis (Fig. 2b). Importantly, three independent experiments were confirmed to yield near identical polymerase profiles (Supplementary Fig. 3). These results demonstrate that the roles of Polε and Polα are conserved between yeasts and humans, although replicon size (the region replicated from a single replication initiation site) is quite different: 30–50 kb in yeasts vs. several hundred kb to 1–1.5 Mb in humans. Interestingly, visual inspection of the profiles shows that the typical enrichment for either leading or lagging strand synthesis was not evident in some areas of the genome. These regions exclusively locate

at heterochromatic late replicating segments (71 sites across the genome, Supplementary Fig. 4). This suggests that DNA replication is regulated differently in these regions (see below).

The profiles of leading and lagging strand polymerases provide two direct and independent measurements of the proportions of replication forks moving either rightward or leftward at each location across the genome. We therefore calculated replication fork direction profiles independently from either the Polε (RFDε) or Polα (RFDα) data and compared these with OK-seq replication directionality data that we calculated directly from the OK-seq sequencing data[20,21] using the identical algorithm (RFDOK; for details see materials and methods). Visual inspection indicates that the overall trends of RFDε, RFDα and RFDOK are highly similar (Supplementary Fig. 5a, b), albeit with the Pu-seq derived data showing lower amplitude peaks. RFDα is more similar to RFDOK than to RFDε (Supplementary Fig. 5c), indicating that the profile of Polα captures additional signatures specific to lagging strand synthesis. Considering the minor differences between leading and lagging strand

synthesis, we established a combined RFD profile from pooled Polε and Polα Pu-seq data. Fluctuation outside of the local trends are notably reduced when compared to RFD$^ε$ or RFD$^α$ (Supplementary Fig. 5a). Thus, combining both leading and lagging strand profiles improves precision as well as resolution of the replication fork profiles (RFD$^{ε|α}$).

## Defining replication initiation regions from polymerase usage

The division of polymerase labour between leading and lagging strands dictates that sites of frequent replication initiation manifest as reciprocal demarcations in Polε and Polα usage. Therefore, we defined an initiation parameter to represent the local activity of initiation events (Fig. 2c). Specifically, we calculate two independent initiation indices (Supplementary Fig. 6a, b), one from the Polε data (Ini$^ε$, where Polε synthesis increases towards the 3′ on the Watson strand and Polε decreases towards the 5′ on the Crick strand) and a second from the Polα data (Ini$^α$, where Polα synthesis decreases towards the 3′ on the Watson strand and Polα increases towards the 5′ on the Crick strand). These two independent indices can be plotted separately (Supplementary Fig. 6a, b) or combined into a more accurate and constrained cumulative initiation index (Ini, Fig. 2c, d). The three replicates of the Pu-seq experiments were used to generate three independent initiation indices (Supplementary Fig. 6c). Positive peaks in the initiation index are interpreted as replication initiation sites and peak height as proportional to the population frequency of initiation. Approximately 12,000 initiation peaks were detected in each replicate (replicate 1: $n = 12,544$, replicate 2: $n = 12,036$, replicate3: $n = 12,311$). The concordance in the peak positions between the replicates increases with increasing values of the initiation index; when all the peaks were taken into account, 67% of all peaks colocalise between replicate 1 & 2 and 50% among all three replicates, whereas these numbers increased to 87% and 80% for the top 50% of initiation peaks. (Supplementary Fig. 6d). Since polymerase profiles were derived from asynchronous cells, this analysis detects initiation events during the entire S phase. Of note, the initiation index can also have negative values, which represents regions where termination of merging forks is frequent (Fig. 2b–d).

Previously mapped initiation sites in the human genome showed that a proportion localised to a few kilobases, whereas other mapped to 'initiation zones' of ~10–50 kb[20,22,23]. In our initiation index profile, the width of positive peaks averaged 34.2 kb (replicate 1: 32.7 kb, replicate 2: 33.1 kb, replicate 3: 36.9 kb) with more than 20% above 50 kb (Fig. 3a). This confirms that initiation events in human cells cluster in zonal regions and that their widths are diverse. The activity of initiation per zone width increases with size, up to approx. 70 kb (Supplementary Fig. 6e) and, when the initiation index is plotted with high-resolution RT data[24], the zones consistently locate at local peaks of RT (Fig. 3b, Supplementary Fig. 7a). We note that initiation zones are present both in mid-late RT regions (e.g. around 41 Mb, 60 Mb in Chr. 3 in Fig. 3b, locations marked by circles in Supplementary Fig. 7a) as well as early replicating regions. These data thus demonstrate that high probability initiation zones are not located primarily in early replicating regions, but exist in mid-late replication regions. This is contrary to the prevailing view that efficient initiation is the predominant determinant of early replicating regions and indicates that late-firing but efficient initiation zones exist across the human genome.

We compared the positions of our initiation zones with those previously detected by various techniques[23,25–27] (Supplementary Fig. 7b). Initiation zones predicted from OK-seq data overlapped the most with Pu-seq initiation zones: compare fractions (vertically) which overlapped with "Pu-seq" in Supplementary Fig. 7b. When we focussed on initiation zones that were concordant in the three Pu-seq replicates, better overlap was seen with the OK-seq predicted initiation zone and the overlapped with bubble-seq and ini-seq data was increased (compare data of "concordant regions" horizontally in Supplementary Fig. 7b). Taken together, these results demonstrate that there is good agreement between Pu-seq and OK-seq initiation zones, both of which

profile the reciprocal pattern of leading/lagging strand DNA synthesis around replication initiation sites.

We also observed 71 regions of Mb length heterochromatic late replicating regions where clear peaks for initiation zones were not evident (Supplementary Fig. 4c). These defined the same regions noted above as having unusual fork direction profiles (Supplementary Fig. 4a). The lack of defined initiation zones was consistent with the reported profile of high-resolution RT data[24], where defined initiation sites were not also observed within these regions (Supplementary Fig. 4d). This likely reflects that replication initiation occurs at random locations within late replicating heterochromatinic regions. This would account for the equal frequency of leftward and rightward moving forks as well as the lack of defined sites of initiation.

## Association of replication Initiation/termination with transcriptional activity

A positional relationship between replication initiation and transcription has been highlighted by multiple studies using different techniques[20,21,23,28]. We therefore analysed how the distribution of transcription units influences the initiation and termination of replication forks in our Pu-seq derived data. The initiation index was aligned at transcription start sites (TSS) and transcription termination sites (TTS). The initiation index score increases in the vicinity of both TSS and TTS (Fig. 3c–e). High levels of gene expression (Fig. 3c) and increased gene length both correlated with increasing initiation index score (Fig. 3d), consistent with published OK-seq data from RPE-1 cells[21]. However, the higher resolution of the Pu-seq derived data shows that the peak of initiation localises ~20 kb upstream of the TSS and ~20 kb downstream of the TTS. We also note that the initiation index shows a negative value throughout gene bodies, consistent with frequent termination in these regions (Fig. 3d, e).

We next aligned the Pu-seq-derived RFD data with TSS and TTS. At TSS and the 5′ regions of genes we observe a significant bias of rightward moving forks. At the TTS and 3′ regions of genes we observe a bias towards leftward moving forks (Fig. 3f, g). Thus, as expected when initiation is biased towards TSS and TTS (and is largely absent from gene bodies), termination is increased within the gene body (Fig. 3h). Notably, these localised initiation and termination patterns are far more apparent for large genes. While the distribution of genes is a factor in the location of initiation and termination zones, our result also demonstrates that there are many genes that are transcriptionally active but do not show replication initiation in the vicinity of their TSS and TTS. Plotting the extent of replication initiation at TSS/TTS against transcriptional activity (Fig. 4a), it is evident that most of the genes associated with replication initiation are transcriptionally active (i.e. area of initiation index > 0 in Fig. 4a). However, transcriptionally active genes are not necessarily associated with replication initiation (i.e. area of log (FPKM) > 0): approximately 20% of genes with > average transcriptional activity show no evidence of initiation in the upstream of TSS. Similarly, plotting the extent of replication initiation at TSS/TTS against gene length, we also observed that there are many large genes which do not show replication initiation in the vicinity of their TSS and TTS (Fig. 4b). Therefore, although transcriptional activity and length of the neighbouring gene is correlated with the extent of initiation (as shown Fig. 3c, d), neither transcription nor gene length are likely to be de a sole determinant of initiation. These data also suggest that many transcriptionally active genes are passively replicated. Furthermore, approximately 42% of loci defined as initiation zones do not overlap with regions upstream of TSS or downstream of TTS (Fig. 4c).

## Replication fork dynamics around transcription start site

It is evident that more forks travel from the initiation sites upstream of the TSS into the gene bodies (i.e. co-directional with transcription) than travel from the gene body through the TSS convergent to the direction of transcription. However, this is not absolute and we

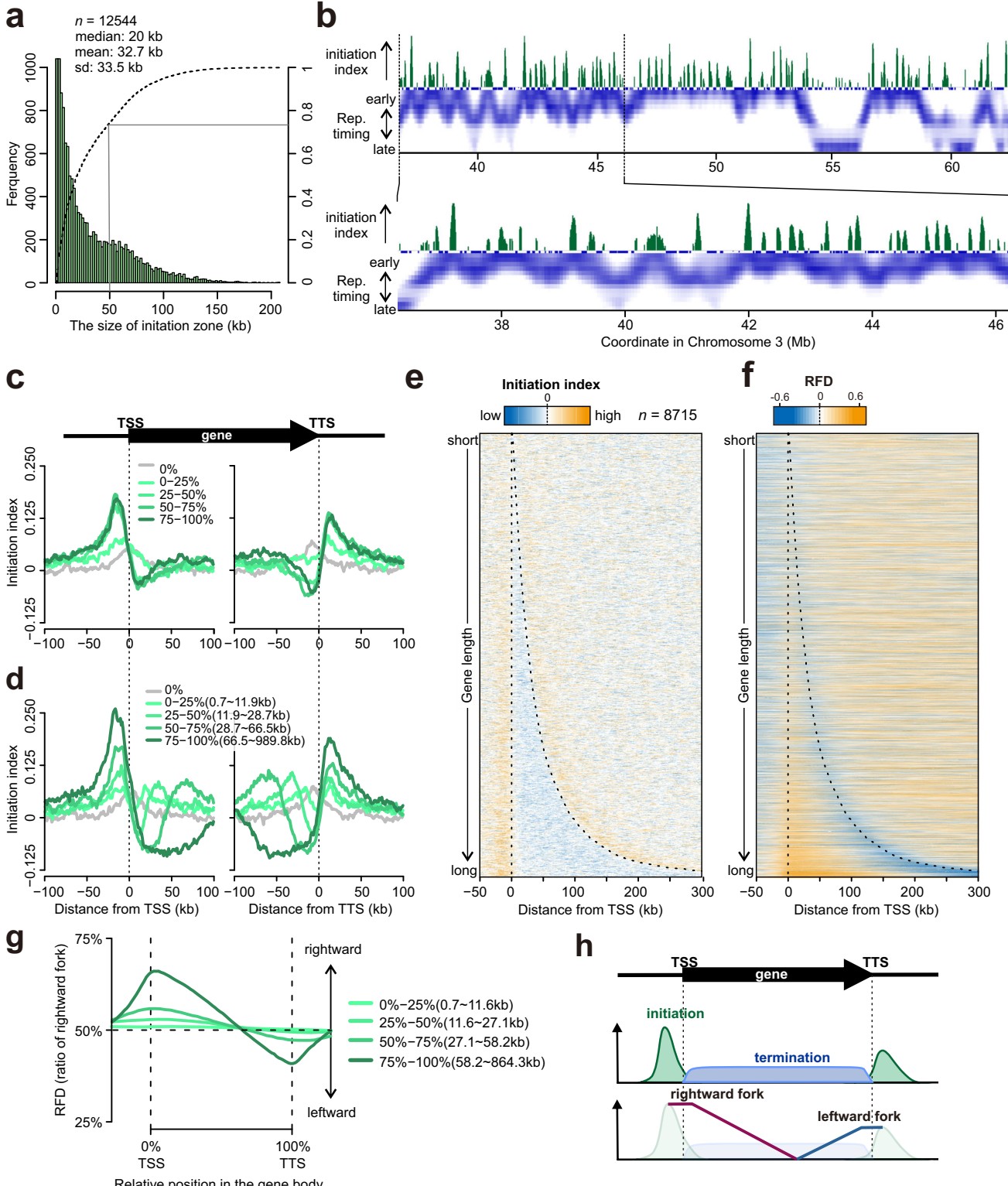

**Fig. 3 | Genomic distribution of initiation sites. a** Distribution of initiation zone widths. **b** High-resolution replication timing data and initiation index plotted for a representative region of chromosome 3. The profile of replication timing for HCT116 cells is from Zao et al.[24]. **c, d** Average initiation index ±100 kb around annotated TSS and TTS in the human genome. Initiation index data are categorised by transcriptional activity (c) or gene length (d). For the gene length analysis only the 50% most transcriptionally active genes were included. **e** Heat map representation of data in panel d sorted by gene length. Broken lines indicate the position of TSS and TTS. **f** Equivalent heat map representation of RFD[ε|α] (from Supplementary Fig. 5a) aligned at TSS and TTS. **g** Average RFD for the relative positions from TSS to TTS for genes scaled to the same arbitrary length. RFD[ε|α] values were converted to rightward fork proportion. For this analysis, only the 50% most transcriptionally active genes were included. **h** Schematic representation of initiation and termination of replication forks as well as fork directionality around a representative gene.

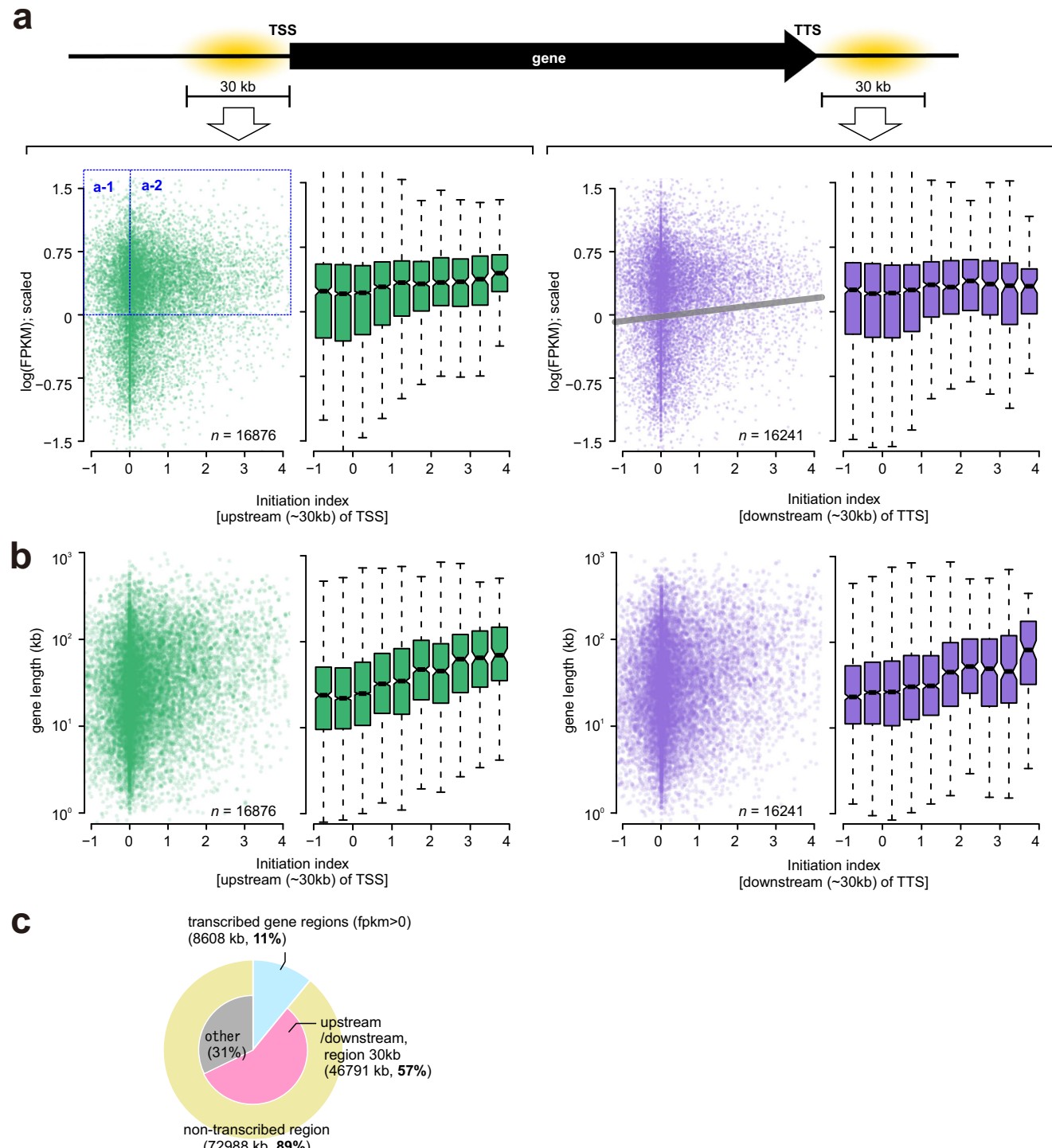

**Fig. 4 | Correlation of initiation and transcriptional activity. a** Top: Schematic of a gene showing TSS and TTS. Bottom: Sum of initiation index within the 0–30 kb region upstream of TSS (left two panels) and downstream of the TTS (two right panels) are plotted against transcriptional activity for each gene, represented as Fragments Per Kilobase of exon per Million mapped reads (FPKM) from an RNA-seq experiment (this study). Sum of initiation index is normalised to Z-score (mean = 0, standard deviation = 1) and Z(0) is subtracted to maintain the original + or – information. Log(FPKM) is normalised to *Z*-score (*n* = 16,876 of TSS and *n* = 16,241 of TTS). **b** Similar to (**a**), sum of initiation index upstream of TSS (left two panels) and downstream of the TTS (two right panels) are plotted against length of genes

(*n* = 16,876 of TSS and n = 16,241 of TTS). **c** Summary diagram of initiation zones overlapped with genes or intergenic regions. Initiation zones were summed (total 81,587 kb) and the ratio of total that overlapped with the following features were calculated; transcribed regions (FPKM > 0, only protein-coding gene): light blue, non-transcribed: yellow, 0–30 kb upstream/downstream region of genes: pink and 'other', which includes intergenic regions and non-gene transcribed regions: grey. Only initiation zones that are concordant in all 3 replicates were analysed. For the boxplots in (**a**, **b**), the horizontal black lines within boxes represent median values, boxes indicate the upper and lower quartiles (25–75%), and whiskers indicate the 1.5× interquartile range.

estimate that, for the top 50% of transcribed genes, this equates to between 66 and 51% co-directional fork and between 49 and 34% of forks that are convergent with transcription, which varies dependent on gene length (Fig. 3g). To separately visualise the dynamics of replication forks dependent on their orientation at any one locus, we calculated a separate 'fork index' for both rightward and leftward moving forks that independently represent their cumulative initiation and termination behaviours. As shown in the formula used to derive the initiation index (Fig. 2d), these profiles can be interpreted as separate 'initiation indices' for rightward and leftward moving forks that originate from non-overlapped polymerase profiles and thus independent parameters. Again, fork initiation and termination are represented by positive or negative values (Fig. 5a). Visual inspection of genome-wide rightward and leftward fork index profiles showed similar genomic profiles that are, as expected, congruent with the initiation index discussed above (Fig. 5b the experimental replicate in Supplementary Fig. 8a).

Aligning rightward and leftward fork indices at TSS allowed us to separately visualise the effect of transcription on forks that are either co-directional (CD) or convergent (CV) with transcription (Fig. 5c, d, Supplementary Fig. 8b). In these figures, leftward moving forks (top-panel) that initiate upstream of TSS – high peak of fork index – move away from the gene and the fork index thus declines slowly. However, leftward moving forks within the gene body are CV with transcription and showed dramatically decreased fork index downstream of TSS. This suggests that head-to-head transcription replication clashes slows fork processivity and increases termination events in this region. In the case of CD forks (rightward moving; bottom-panel) the fork index profile at TSS is more complex: two low positive peaks are evident. We interpret this as a combination of initiation and termination: i.e. a negative signal (fork termination) at 0–20 kb upstream of TSS is embedded within a strong positive signal (fork initiation). Visualising the fork index of individual regions by heatmap, we observed that the fork index in these regions is heterogenous, reflecting a mixture of fork termination and initiation (Fig. 5e, Supplementary Fig. 8c). Compared to CV forks, CD forks show more termination events and their extent is higher at upstream of TSS. Thus, the ensemble signal of fork termination and initiation manifested in fork index does not necessary represent the profile of all individual loci. It should also be noted, as discussed above, that not all transcriptionally active genes are associated with an increase in initiation upstream of TSS (Fig. 4a). In the transcriptionally active genes (> average) which do not show an associated increase of replication initiation at TSS (represented by region "a-1" in Fig. 4a), the fork index for CD forks drops sharply below zero at and immediately upstream of TSS (Supplementary Fig. 8d), indicating that forks exclusively terminate in this region. In contrast, the fork index profile of genes that show association of initiation with the TSS resembles that of all active genes (represented by region "a-2" in Fig. 4a). Combinedly, these data showed that fork termination commonly occurs in the vicinity of TSS of active genes.

The combined pattern of CV and CD forks is consistent with the trend for fork initiation in both orientations ~20 kb upstream of TSS, with the processivity of leftward moving (CV) forks being reduced immediately downstream of TSS (Fig. 5d, top) and the processivity of rightward moving (CD) being reduced immediate upstream of TSS (Fig. 5d bottom). We confirmed this pattern is almost abolished without transcription (Supplementary Fig. 8e). Taking gene length into account demonstrates that the effect of termination is largely independent of gene length and manifests throughout the length of the transcription unit (Fig. 5f, Supplementary Fig. 8b). This tendency is contrary to fork initiation, which increase with gene length. During transcription initiation RNA Polymerase II (RNAPII) promoter-proximal pausing is enriched in the immediate vicinity of TSS[29]. Thus RNAPII, directly or indirectly, likely causes an impediment to fork progression. In contrast, as predicted by the initiation index around TTS, both

initiation and termination events are evident and fork orientation (co-directional or convergent) did not influence the profiles.

## Local genomic features within initiation zones

We next investigated if specific sequence-based or chromatin-based features correlate with replication initiation zones. By using the chromosomal coordinate of the highest initiation signal within each initiation zone (yellow vertical lines in Fig. 2c bottom and Supplementary Fig. 6c) we examined if specific genomic elements are enriched at initiation zone peaks. GC skew, AT skew and CpG islands were not enriched. Potential guanine quadruplex (G4) structures were modestly enriched at peaks of initiation index (Supplementary Fig. 9a). For chromatin features, the H2AZ histone variant and, to a lesser extent, trimethylated H3K27 (H3K27me3) were enriched at initiation zone peaks (Supplementary Fig. 9b). In contrast, trimethylated H3K36 (H3K36me3) tended to be excluded from and moderately enhanced in flanking regions, likely because this histone mark is associated with gene bodies[29]. We also noted that H3K4me3 was enriched ~ 20 kb either side of the initiation zone peaks, consistent with its enrichment at TSS[29] (Supplementary Fig. 9c). Furthermore, analysing genomic status based on multiple chromatin profiles by using ChromHMM algorism[30], initiation zones and TSS/TTS were shown to associate with distinct chromatin states (Supplementary Fig. 9d). In summary, the chromatin features H2AZ and H3K27me3 positively correlate specifically with initiation sites, while H3K36me3 shows a minor anti-correlation.

To establish which of the features discussed above correlate best with initiation activity we partitioned each chromosome into 10 kb-bins and used principal component analysis (PCA) to deconvolve genomic features in relation to replication initiation (Supplementary Fig. 9e). This revealed that chromatin-based features (H2AZ and H3K27me3) correlated more closely with replication initiation than G4. As expected, H3K36me3, which is deposited in transcribed regions in an RNAPII-dependent manner (thus marking gene bodies), positioned oppositely to initiation. These data suggest that, combinatorially with transcription, chromatin status is crucial for shaping replication initiation.

## Uncoupling of leading and lagging strand polymerases

Having two independent datasets that represent leading (Polε) and lagging (Polα) strand synthesis offers the opportunity to examine how well coupled DNA synthesis is throughout the genome. As expected, Polε and Polα usage on the leading and lagging stands respectively, for forks moving in the same direction, is notably similar. However, reproducible differences in Polε and Polα profiles are detected (e.g. around 26 Mb on chromosome 6 in Fig. 6a, experimental replicate: Supplementary Fig. 10). We interpret this as evidence of uncoupling between leading and lagging strand polymerases[31]. To quantify this, we calculated a separate coupling index for rightward and leftward moving forks to represent the bias toward either Polε or Polα usage (Fig. 6b). If both leading and lagging strand synthesis contribute equally to replication of the duplex DNA, coupling index = 0. Positive values of coupling index represent a bias toward leading strand polymerase (Polε) while a negative value represent a bias toward lagging strand polymerase (Polα).

Across the majority of genomic regions, the coupling index remains close to zero, but at some loci it reproducibly deviates by +/−0.25, suggesting up to 25% biased usage of Polε or Polα occurs relatively frequently across the genome (Fig. 6c, d). We observed a significant correlation between biological replicates 1 and 2 (Supplementary Fig. 11a). Interestingly, we observed a reciprocal pattern for coupling index between rightward and leftward moving forks (Fig. 6d, Supplementary Fig. 11b). This indicates an opposite bias for forks moving in the two directions is present. For example, at the histone-encoding gene clusters on chromosome 6, usage of Polε was over-represented compared Polα in the rightward forks and the opposite trend was observed in leftward forks (Fig. 6c, Supplementary Fig. 10).

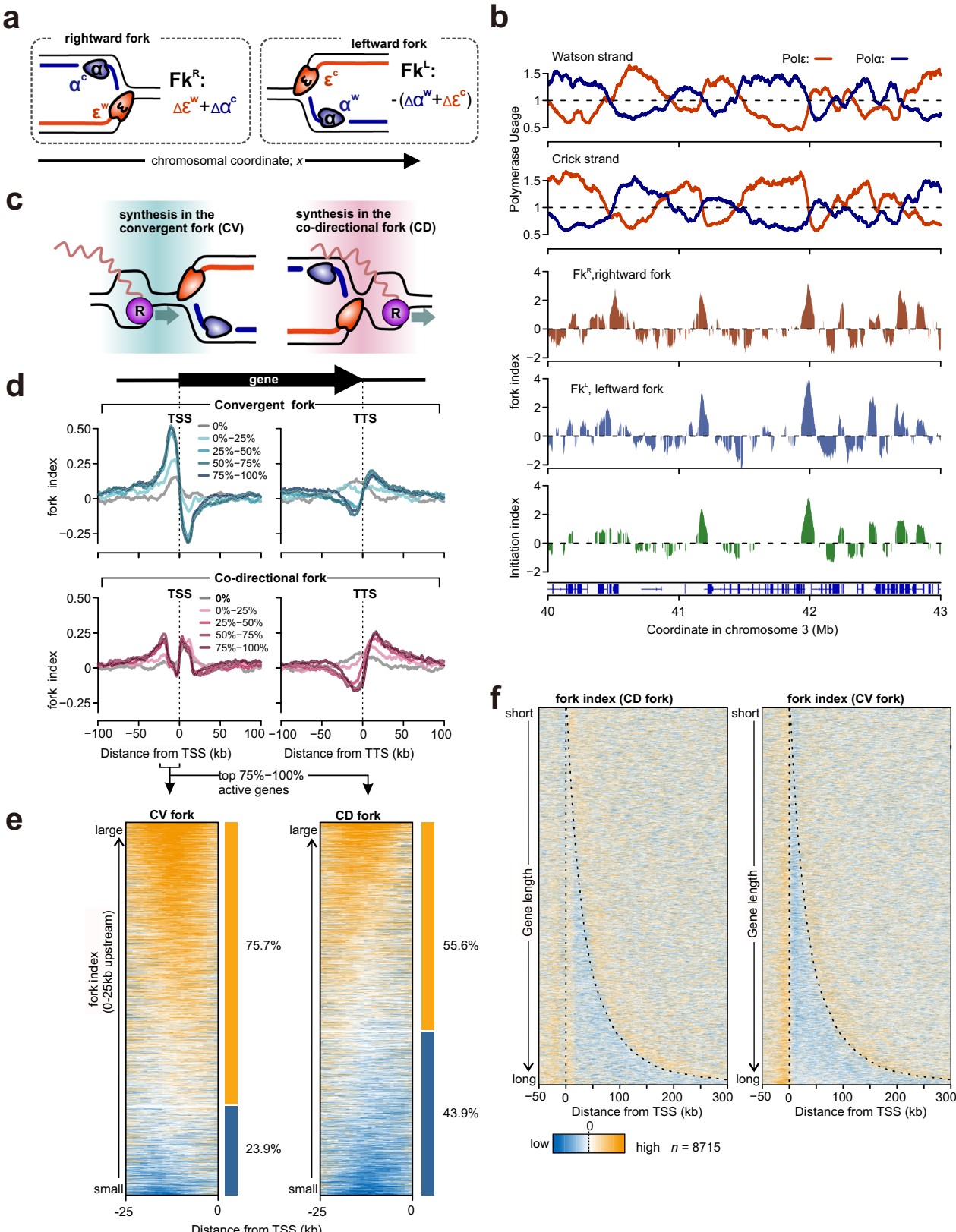

This inverse correlation is conserved across the genome (replicate 1: $r = -0.403$, replicate 2: $r = -0.354$, replicate 3 $r = -0.521$, Fig. 6d, Supplementary Fig. 11b). We next examined if these coupling index fluctuations correlated with transcription. By aligning coupling index of co-directional (CD) and converging forks (CV) at TSS sites we observed that the coupling index of CD forks increased within gene bodies,

whereas the coupling index of CV forks decreased (Fig. 6e, f). This likely reflects the orientation of DNA polymerase movement and transcription: synthesis by forks moving in the same direction of RNAPII encounter problems that result in a bias toward Polε (suggesting lagging strand synthesis is impaired), while synthesis by forks moving in the opposite direction to RNAPII result in problems that

**Fig. 5 | The profiles of rightward and leftward forks. a** Definition of fork indices of rightward and leftward forks (Fk$^R$ and Fk$^L$, for further details see materials and methods). '$\Delta$' indicates the differential between neighbouring bins, e.g. at location $x$, $\Delta\varepsilon(x)$ is defined as $\varepsilon(x+1)$-$\varepsilon(x)$. **b** Profiles of polymerase usage (top), fork index (middle) and initiation index (bottom) for a representative region of chromosome 3. **c** Schematic representation of transcription and replication conflicts for convergent (CV) or co-directional (CD) forks. **d** Averaged fork index ±100 kb around annotated TSS and TTS in the human genome. Data for the fork indices of CV and CD forks are categorised by transcriptional activities. **e** Heat map representation of CV and CD fork index at 25–0 kb upstream regions of active TSS (the 75–100% category in **d**). Data were sorted by the total value of fork index in each region. Yellow and blue bars indicate regions with positive or negative values, respectively. **f** Heat map representation of fork indices of CV and CD forks sorted by gene length. Broken lines indicate the position of TSS and TTS. For this analysis only the 50% most transcriptionally active genes were included.

result in a bias towards Polα (suggesting that leading strand synthesis is impaired). Consequently, DNA synthesis on the non-transcribed strand, either by lagging stand synthesis at CD forks or leading stand synthesis at CV forks, is shown to be impaired (see depiction of CD and CV forks in Fig. 6e, further discussed below).

The genome wide distribution of coupling index variation tends toward lower values (Fig. 6d, Supplementary Fig. 11b). This suggests that leading strand synthesis is generally more susceptible to spontaneous impediment than lagging strand synthesis. The affected regions are not uniformly dispersed across the genome and tend to favour particular chromosomes (Fig. 7a). To identify regions associated with this phenomenon, we statistically identified 27 regions as coupling index outliner loci, where either rightward or leftward coupling index values diverged significantly from the population of chromosomal data in two biological replicates (Table 1). Among these, 16 loci (59.2%) were associated with large genes. In all but one of these the coupling index was notably low (coupling index < −0.3) for forks converging with transcription. One explanation of these results is transcription stress due to head-to-head collisions becomes intense in a subset of large genes and consequently polymerase uncoupling occurs frequently due to an impediment of leading strand DNA synthesis. However, it should not be excluded that an alternative engagement of other polymerases, such as Polδ during replication restart[32,33] causes the low CI values.

In three of the uncoupling regions (Table 1: Chr. 3 60–61 Mb, Chr. 16 78.7–78.9, Chr. 21 19.2–19.6) coupling index values were significantly negative for both rightward (CD) and leftward (CV) moving forks. This indicates perturbation of leading strand DNA synthesis on both strands of the duplex (Fig. 7b, c). Two of these three loci are positioned within the FRA3B and FRA16D common fragile sites (CFSs) that are highly expressed in many cell lines, including HCT116[34]. The local pattern low coupling index loci did not fully match with those of replication initiation or termination zones and thus fork dynamics does not solely account for low coupling index in these regions, suggesting that specific impediments to leading strand polymerases occur at these regions. Comparing our data with published End-seq experiments in HCT116[35], the low coupling index loci do not overlap with hotspots of double strand breaks (DSBs). Thus, we propose that uncoupling of replicative polymerase is separate feature of at least a subset of CFS and can contribute to chromosome rearrangement in a manner independent of DSBs.

## Discussion

By locating the positions of incorporated rNMP by mutated DNA polymerases in human HCT116 cells we have characterised genome-wide usage of the replicative polymerases Polε and Polα. The profiles of Polε and Polα usage on either the Watson or Crick strands are strikingly reciprocal. While Polα primes both leading and lagging strand synthesis, the number of lagging strand priming events vastly exceed that of the leading strand. Thus, our data demonstrate that human Polε and Polα contribute to leading and lagging strand DNA synthesis, respectively, across the genome. These results establish that the roles of Polε and Polα, and by implication Polδ, are conserved between human and yeasts[5–7]. Our attempts to characterise Polδ were not successful because the POLD1-L606G or POLD1-L606M mutants (predicted to elevate rNMP-incorporation) could not be isolated. Our approach requires transient degradation of RNASEH2A, which could potentially perturb

replication dynamics by causing replication stress. However, during the 48 h in which we inactivate RNASH2A, Chk1 is not phosphorylated, BrdU incorporation is not perturbed and we see no global increase in DNA/RNA hybrid structures. We thus conclude that transient RNASEH2A degradation does not influence the global trend of DNA replication.

Our analysis of replication initiation identified approximately 12,000 sites with a positive initiation index in each biological replicate, comparable of the numbers identified by Ori-SSDS for mouse cells (11,000–13,000)[36]. The average inter peak distance is 230 kb, slightly larger than previous estimates (160–190 kb)[37]. The average width of the zones showing a positive initiation index was (mean 34 kb, max. 235 kb), comparable to OK-seq[20] (HeLa cells: mean 31 kb, max. 143 kb, GM06990 cells: mean 34 kb, max. 254 kb) and optical replication mapping[38] (mean 32 kb, max. 189 kb). These data confirm that initiation in human cells is zonal. Consistent with previous reports, we observed a correlation between replication initiation and TSS/TTS of transcribed genes. However, we also demonstrate that an appreciable fraction of active genes are not associated with initiation at TSS and/or TTS. Currently we have not identified common features for genes without and with associated initiation zones and speculate that higher order of nuclear structures may need to be taken into account to identify such features. We note that, during preparing of this manuscript, cohesion-mediated looping has been proposed as a key factor determining initiation zones[39]. An examination of chromatin features identified H2AZ and, to a lesser extent H3K27me3, as correlating with initiation. Histone modifications directly associated with transcription, such as H3K4me3 (TSS associated[29]) and H3K36me3 (gene body associated), showed separate and distinct profiles that likely reflect the proximity of these modifications at TSS or gene bodies and hence unlikely to be causative correlations. We speculate that H2AZ and H3K27me3 contribute to configuring an accessible local environment for initiation zones, in which H2AZ and/or other chromatin features recruit ORC or pre-RC components[40].

Because we could separately analyse rightward and leftward fork dynamics, we have established that both co-directional and convergent forks are impaired around TSS – most likely when they encounter paused RNAPII, which is frequent around transcription initiation sites. This observation is consistent with a previous report showing a delay to replication kinetics at TSS[41]. An accumulation of RNAPII pausing has been reported as a phenomenon associated with cancer-prone situations[42,43]. The consequent increase in replication-transcription conflicts, such as we demonstrate here, may underlie this association. We note that the impairment to fork dynamics calculated for a single direction (rightward and leftward fork index) does not appear in our initiation indices (Ini$^\varepsilon$, Ini$^\alpha$ and the combined index, Ini) because initiation indices are defined to describe fork dynamics across both orientations. Thus, by establishing the effects of transcription on both the initiation index and fork index, our analysis separately detects the impediment to fork progression at TSS or the termination of two merging forks, which we show occur at different preferential locations relative to genes.

Having strand specific data for two independent polymerases allowed us to define a fork coupling index (see Fig. 6b) that quantifies how well leading and lagging strand synthesis are coupled during fork progression. Surprisingly, a reproducible bias toward either Polε (leading) or Polα (lagging) strand polymerases is relatively common. The fluctuation of the coupling index for human cells appear much

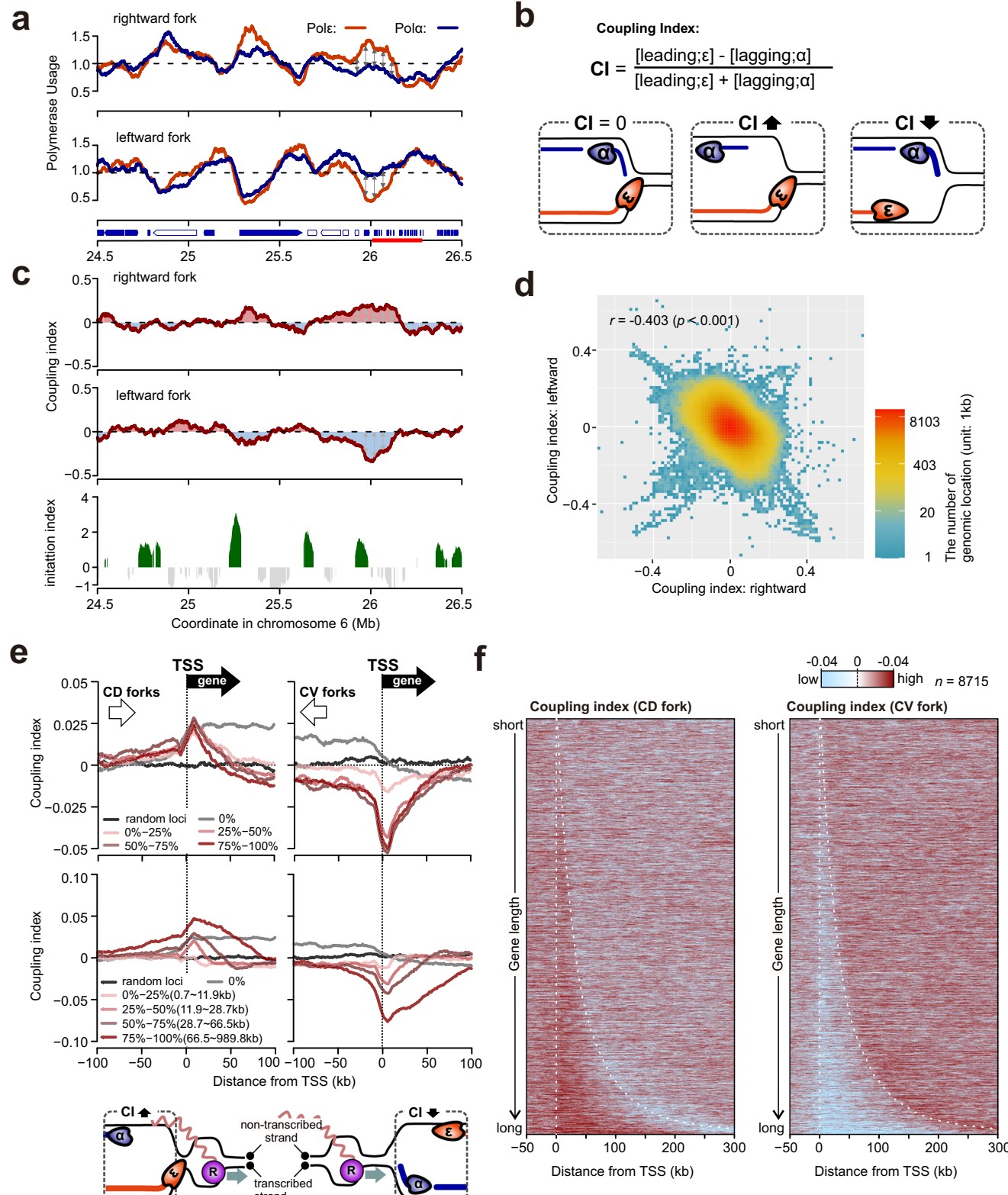

**Fig. 6 | The uncoupling of leading and lagging strand DNA synthesis. a** Profiles of polymerase usage for a region of chromosome 6. Arrows on the horizontal axis indicate active genes (blue-filled) and inactive genes (unfilled). The red line on the horizontal axis indicates a cluster of histone-encoding genes. **b** Definition of coupling index (CI, for details see materials and methods). **c** Top: CI of rightward moving forks. Middle: CI of leftward moving forks. Bottom: the initiation index of the same region for comparison. **d** Correlation of CIs of rightward and leftward forks. *r* represents Pearson correlation coefficient and *p* shows its statistical significance (one-tailed *t* test). **e** Averaged coupling index ±100 kb around annotated TSS in the human genome for CV and CD forks categorised by transcriptional activities (top) or gene length (middle). For gene length analysis, only the 50% most transcriptionally active genes were included. Bottom: depiction of polymerase uncoupling in CD and CV forks, **f** Heat map representation of data in panel e. Data were sorted by gene length. Broken lines indicate the positions of TSS and TTS.

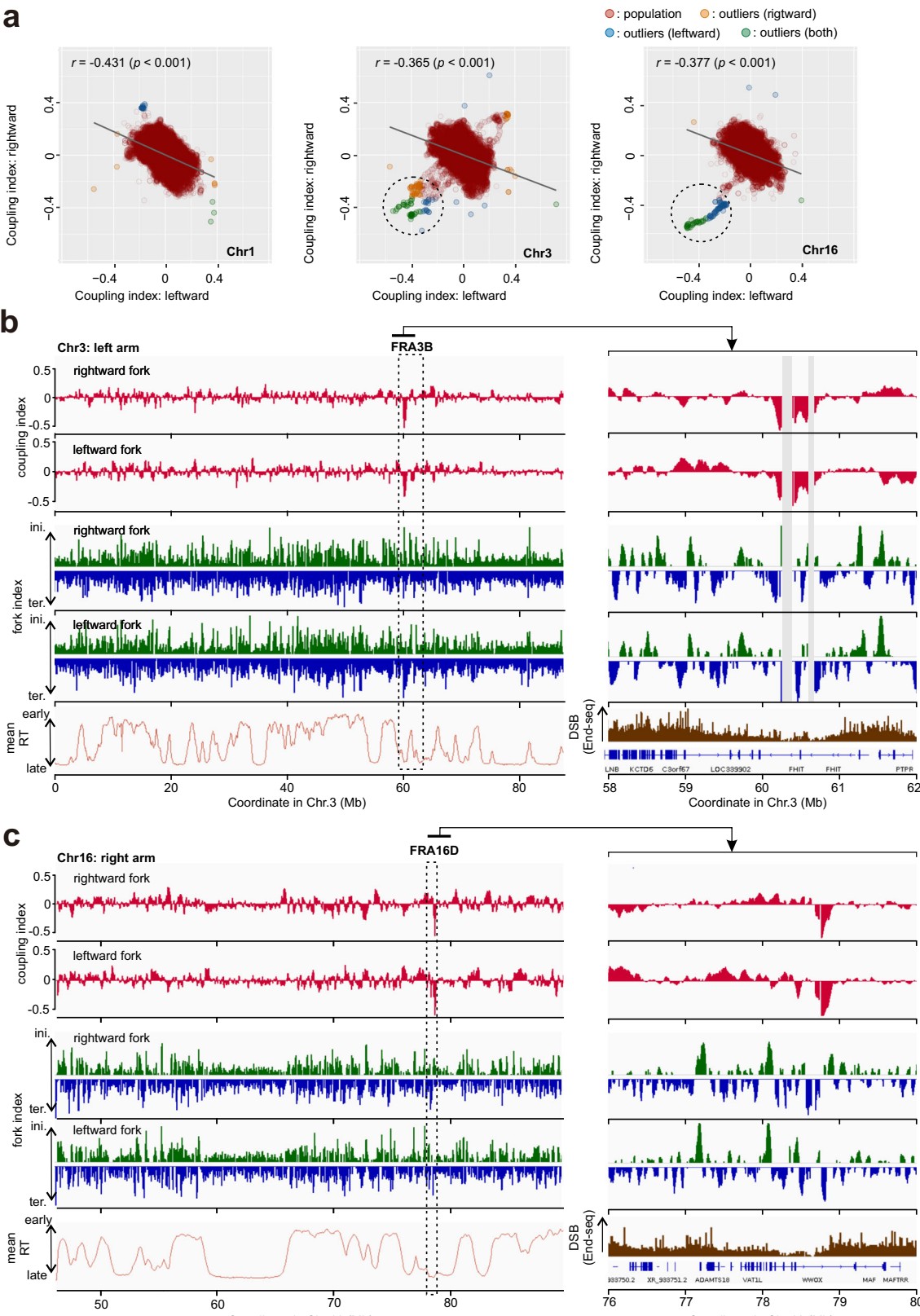

**Fig. 7 | Polymerase uncoupling at common fragile sites FR3B and FRA16D.**
**a** Correlation of coupling index of rightward and leftward moving forks presented by chromosome. Chromosomes 1, 3 and 16 are shown. The dashed circles highlight data points dispersed from the bulk of population. These outlier data points were identified by Smirnov-Grubbs test. *r* represents Pearson correlation coefficient and *p* shows its statistical significance (one-tailed t-test). **b** Top: profiles of coupling index for rightward and leftward forks. Middle: fork index profile. Bottom: mean replication timing (mean RT) or DSB data for the FRA3B region on chromosome 3. Grey bars mask regions where the profiles are void due to a low count of sequencing reads (see methods). **c** The equivalent data for the FRA16D region of chromosome 16. Replication timing data for HCT116 cells are derived from Zao et al. [24]. Data of DSBs from End-seq are derived from Tubbs et al. [35].

**Table 1 | Coupling index outliner loci**

| Location | | Neighbouring gene | | | Coupling of replicative DNA polymerases | |
|---|---|---|---|---|---|---|
| Chrom. | Position (Mb) | Name | Size (kb) | direction | Coupling index <0 (leading <lagging)* | Coupling index > 0 (leading > lagging)* |
| 2 | 235.4–236.0 | AGAP1 | 638 | rightward → | ← leftward fork | – |
| 3 | 60–61 | FHIT | 1520 | ← leftward | ← leftward & rightward → fork | – |
| 3 | 114.4–114.6 | ZBTB20 | 852 | ← leftward | – | rightward fork → |
| 3 | 188.1–189.1 | LPP | 757 | rightward → | ← leftward fork | – |
| 4 | 19.5–20.0 | – | | – | rightward fork → | – |
| 4 | 49.1–49.2 | – | | – | rightward fork → | – |
| 5 | 36.8–37.1 | NIPBL | 190 | rightward → | ← leftward fork | – |
| 5 | 131.5–131.8 | FNIP1 | 155 | ← leftward | rightward fork → | – |
| 9 | 123.6–124.0 | DENND1A | 550 | ← leftward | rightward fork → | – |
| 10 | 38.4–38.6 | – | | – | ← leftward fork | – |
| 10 | 41.8–42.1 | – | | – | ← leftward fork | – |
| 10 | 87.2–87.4 | NUTM2A-AS1 | 139 | ← leftward | rightward fork → | – |
| 13 | 60.0–60.2 | DIAPH3 | 498 | ← leftward | rightward fork → | – |
| 13 | 77.2–77.4 | MYCBP2 | 282 | ← leftward | rightward fork → | – |
| 13 | 98.1–98.3 | FARP1 | 307 | rightward → | ← leftward fork | – |
| 15 | 41.0–41.2 | INO80 | 137 | ← leftward | rightward fork → | – |
| 16 | 34.6–34.8 | – | | – | rightward fork → | – |
| 16 | 78.7–78.9 | WWOX | 1113 | rightward → | ← leftward & rightward → fork | – |
| 17 | 21.8–22.0 | – | | – | leftward fork | – |
| 17 | 26.7–26.9 | – | | – | leftward fork | – |
| 18 | 49.2–49.6 | DYM | 418 | ← leftward | rightward fork → | – |
| 20 | 31.0–31.3 | – | | – | rightward fork → | – |
| 21 | 7.9–8.0 | – | | – | rightward fork → | – |
| 21 | 10.6–10.8 | – | | – | rightward fork → | – |
| 21 | 19.2–19.6 | – | | – | ← leftward & rightward → fork | – |
| 22 | 40.4–40.7 | MRTFA | 226 | ← leftward | rightward fork → | – |
| 22 | 47.1–47.2 | TBC1D22A | 433 | rightward → | rightward fork → | – |

*As shown in Fig. 7a, Smirnov-Grubbs test were used to determine the genomic locations of which coupling index of either rightward or leftward forks is outliers in those of the whole chromosome.

greater than that previously observed in yeasts[5,12] and, unlike in yeasts, where polymerase bias is associated with initiation or termination, coupling index fluctuations are prominent in regions not associated with these phenomena. Molecular events that might generate coupling index fluctuation include the inhibition of DNA synthesis to generate strand-specific polymerisation defects or DNA synthesis by other polymerases. Thus, coupling index fluctuation likely reflect multiple mechanisms that can occur either within ongoing replication forks or after fork passage or collapse.

Studies using *E. coli* or budding yeast proteins or *Xenopus* extracts[44–46] showed that, upon uncoupling of the helicase from leading strand polymerase, the helicase slows to encourage recoupling. In vitro analysis using budding yeast proteins suggests that Polδ recouples synthesis. In our analysis, this would manifest as a negative coupling index (bias towards Polα). Alternatively, if DNA polymerase usage at uncoupled regions is determined by noncanonical polymerases not intrinsic to the fork, the coupling index would fluctuate toward either negative or positive values, depending on whether leading or lagging stand synthesis was influenced. While noncanonical polymerase usage has mainly been associated with DNA damage tolerance, there is increasing evidence for such phenomena under unperturbed conditions, particularly in mammalian cells[47–51]. Thus, our data raise the important question of how extensively the previously unanticipated level of the plasticity of DNA polymerase usage underpins replication of the human genome. Of note, we demonstrate that inhibition of the leading strand polymerase is particularly evident in FRA3B and FRA16D in the absence of exogenous replication stress.

Consistent with clashes between replication and transcriptional machineries, we observed increased coupling index fluctuations in gene bodies, implying transcription-replication conflicts cause frequent polymerase uncoupling. Separating out the effects on convergent and co-directional forks, the block to DNA synthesis tends to manifest where the non-transcribed strand is used as template DNA. One potential explanation is that, when DNA is transcribed, R-loop structures can form behind RNAPII leading to transient regions of ssDNA on the non-transcribed strand (Supplementary Fig. 12). ssDNA is chemically less stable than dsDNA and more susceptible to damaging agents and some DNA-modifying enzymes[52]. Indeed, transcription-coupled 'damage' on the non-transcribed DNA strand has been shown to cause increased mutational load in some cancer cells[53]. It is also of note that transcription coupled repair eliminates DNA damage specifically on the transcribed strand. As a result, DNA damage on the non-transcribed strand may become relatively more influential in perturbing DNA synthesis (Supplementary Fig. 12). Whatever the underlying causes, this transcriptional effect is particularly manifested in large genes, which may reflect the fact that transcription must proceed beyond the end of S phase[54]. Another possibility is that the accumulation of positive supercoiling ahead of, and negative supercoiling behind, RNAPII not only pauses replication forks but also enhances the level of transcription-induced blocks, perhaps by inducing excessive fork rotation, which previous studies in budding yeast have found to generates post-replicative stress[55].

In summary, Pu-seq in human cells provides a powerful and straightforward methodology to explore DNA polymerase usage and

replication fork dynamics. We show here that data produced from Pu-seq are highly consistent with those from other methods such as high resolution repli-seq and OK-seq. A strength of Pu-seq is that the two independent DNA polymerase datasets are generated, which provides four separate measures of replication dynamics, allowing predictions of replication fork movement, initiation and termination with great accuracy. Exploiting this we show that transcription influences circa 60% of initiation events and contributes significantly to replication fork uncoupling and subsequent genome instability. Pu-seq will thus provide a useful tool for examining DNA replication by manipulating specific genetic changes in the HCT116 cell lines we have developed. Although we have currently only applied Pu-seq to a single metazoan cell line, HCT116, which is derived from colon cancer and hypermutable, we anticipate that, by constructing the system in different cell lines harbouring distinct developmental or genetic backgrounds, the methodology will allow detailed analysis of the intrinsic flexibility of DNA replication

## Methods

### Cell culture

All cell lines are derived from HCT116 cells (ATCC, #CCL-247) and are listed in Supplementary Table 1. Cells were cultured in McCoy's 5 A, supplemented with 10% FBS (Gibco, #10437-028), 2 mM L-glutamine, 100 U/ml penicillin and 100 µg/ml streptomycin at 37 °C in 5% $CO_2$. IAA (Nakarai tesque #19119-61), 5-Ph-IAA (BioAcademia #30-003) and auxinole (BioAcademia #30-001) were dissolved in DMSO to create a 500 mM stock solution (stored at −20 °C) and further diluted with media to an appropriate 25 x concentration and added directly to the culture medium to achieve the working concentration (IAA 500 µM, 5-Ph-IAA 400 nM, auxinole 100 µM). Doxycycline was dissolved in water to create a 1 mg/ml stock solution (stored at −20 °C) and added directly to the culture media to achieve working solution (0.2 µg/ml).

### Generation of polymerase mutants

Cas9 protein (Alt-R® S.p. HiFi Cas9 Nuclease V3, #1081060) and tracrRNA (Alt-R® CRISPR-Cas9 tracrRNA, #1072533) and crRNA (Alt-R® CRISPR-Cas9 crRNA, custom production, Supplementary Table 1) were purchased from IDT (Integrated DNA Technologies, USA). Guide RNA (gRNA) was formed by mixing equimolar amounts (50 µM) of tracrRNA and crRNA in duplex buffer (IDT), heating to 96 °C for 5 min, and cooling on the benchtop to room temperature. 61 µM of Cas9 and gRNA were mixed at a ratio of 2:3 and incubated for 0.5 −1 hr. Following Ribonucleoprotein (RNP) formation, 1.5 µl of RNP and 1.5 µl of 36 µM single stranded oligodeoxynucleotide (ssODN) template (listed in Supplementary Table 1) was added into $1 \times 10^5$ cells in 12 µl of R buffer in Neon® transfection system 10 µl Kit (Invitrogen, #MPK1096) and the cell suspension applied to Neon® transfection system with 10 µl Neon tips. Following electroporation cells were immediately suspended into 500 µl of media and Alt-R™ HDR Enhancer V2 (IDT, #10007921) to achieve 20 µM was added. 3 day after transfection, 300–3000 cells were plated in 10 cm dishes for colony formation. 96 single colonies were picked into wells of a 96 well plate. After replicating the clones into two 96 well plate cells were incubated for 2–4 days. The cells on one plate was subjected to genotyping by PCR and those on the other plate was stored at −80 °C with Bambanker DIRECT medium (Nippon Genetics, CS-06-001).

### Generation of mAID-Clover–tagged RNASEH2A cell line

Cells were transfected with CRISPR–Cas9 and donor plasmids (Supplementary Table 1) using FuGENE HD Transfection Reagent (Promega, #E2311) in a 6-well plate following the manufacturer's instructions. Two days after transfection, cells were plated in 10 cm dishes and selected with antibiotics. Selected clones were isolated and confirmed as previously described[56].

### The degradation of RNASEH2A cell line in AID or AID2 system

For the AID system, the +TetOsTIR1 mAID-Clover–tagged RNASEH2A cells (Supplementary Table 1) were treated with doxycycline and auxinole[56] for 24 hrs to induce OsTIR1 without background degradation and this medium was replaced with medium containing doxycycline and IAA. The cells were incubated for a further indicated period. For AID2 system, where OsTIR1(F74G) is constitutively expressed, +OsTIR1(F74G) mAID-Clover–tagged RNASEH2A cells (Supplementary Table 1) were treated with 5Ph-IAA for the indicated period.

### DNA extraction, alkaline treatment and library preparation

In total, $2 \times 10^7$ cells were harvested by centrifugation and genomic DNA was prepared using Blood & Cell Culture DNA Midi Kit 100/G Genomic-tips (Qiagen #13343). To examine alkaline degradation 3 µg of DNA was treated with 0.3 M NaOH at 55 °C for 2 hr in 15 µl. The reaction was stopped by adding 3 µl of 1 M Tris-HCl (pH7.5). 1 µl of this solution was subjected to TapeStation RNA ScreenTape Analysis (Agilent #5067-5576, #5067-5577, #5067-5578) to detect ssDNA. Visualisation of DNA fragment patterns and quantification of DNA < 2.0 kb were performed using TapeStation Software (Agilent). For library preparation 25 µg of genomic DNA was alkali treated in 0.3 M NaOH at 55 °C for 2 h, then loaded onto a 1.5% agarose gel and run for 1 h 40 min at 100 V. The gel was stained with acridine orange (final concentration 5 µg/ml) for 2 h at room temperature with gentle shaking followed by overnight destaining in water. Fragments of 300–2000 bp were excised from the gel and isolated with a gel-extraction kit (Macherey-Nagel, NucleoSpin Gel and PCR Clean-up, #740609). Library preparation was performed as previously described[5,57]. Libraries were 150-bp paired-end (PE) sequenced on an Illumina Hiseq X platform (Macrogen, Tokyo, Japan).

### Cell growth assay

Cells were plated onto a 6-well plate and cultured in the presence or absence of 400 nM 5-Ph-IAA/500 µM IAA. Growth curves were obtained by measuring the cell count after trypsinising cells at the indicated time points. Relative cell density was calculated by taking the density recorded at time 0 as 1.

### Dot blot assay

Detection of BrdU was performed as described previously[58]. The concentration of genomic DNA was adjusted to 0.4 µg/µl. After heat-denaturation and snap-cooling, 2 µl from this solution (0.8 µg of DNA) were spotted on the nylon membrane (Amersham™ Hybond™-N+, Cytiva, RPN1210B) and subjected to ultraviolet crosslinking in a UV Crosslinker (UBP, CX-2000) at $1.2 \times 10^5$ µJ. Subsequently, the membrane was equilibrated in phosphate-buffered saline (PBS) containing 0.1% Tween-20 (PBST), blocked with 5% non-fat milk in PBST for 30 min and probed with a monoclonal BrdU antibody (dilution 1:500, BD Biosciences 347580) in PBST + 0.5% milk overnight at 4 °C. After washing and addition of secondary antibody (dilution 1:10000, anti-mouse HRP conjugate in PBST + 0.5% milk), the blot was developed using Western Lightning Plus-ECL (Perkin Elmer, NEL104001EA) and the signals were imaged using a ChemiDoc Touch MP system (BioRad). The signal intensity of BrdU dots was quantified using Image Lab software (BioRad).

To detect DNA/RNA hybrids, 1 µg of genomic DNA was treated with E. coli RNase H (NEB, M0297S) at 37 °C for 1 hr or mock treated in the presence of RNase A (0.1 mg/µl, Merck 9001-99-4) in 10 µl. 2 µl from this solution (200 ng of DNA) was spotted on the nylon membrane (Hybond N+, GE Healthcare) and subjected to ultraviolet crosslinking in a UV Crosslinker (UBP, CX-2000) at $1.2 \times 10^5$ µJ. The same immunostaining procedure as BrdU detection was performed except use of a monoclonal S9.6 antibody (dilution: 1:2000, abcam, ab234957) as primary antibody.

## Western blotting

Approx. $2 \times 10^5$ HCT116 cells were lysed in RIPA buffer (25 mM Tris-HCl pH7.6, 150 mM NaCl, 1% NP40, 1% sodium deoxycholate, 0.1% SDS) and sonicated (30 s on and 30 s off x 15 cycles) with Bioruptor II (BMBio). After centrifugation, the supernatant was mixed with 4 x bolt LDS sample buffer (Invitrogen™ B0007) supplemented with 10 x bolt sample reducing agent (Invitrogen™ B0004) before incubation at 70 °C for 10 min. After SDS-PAGE with bolt 4–12% Bis-Tris plus gels, proteins were transferred to the immune-Blot PVDF membrane (BioRad #1620177). After blocking with PBST + 5% milk, the membrane was incubated with a primary antibody at 4 °C overnight in PBST + 0.5% milk and subsequently incubated with a secondary antibody at room temperature for 1 hr PBST + 0.5% milk. Detection was performed using western lightning plus-ECL (Perkin Elmer, NEL104001EA) and images were acquired with a ChemiDoc Touch MP system (Bio-Rad). Antibody used: anti-RNASEH2A (dilution 1:2000, Bethyl Laboratories® A304-149A), anti-phospho-Chk1 (dilution 1:1000, Cell Signalling Technology #2348), anti-tubulin (dilution 1:5000, Sigma-Aldrich T5168), anti-mouse IgG HRP conjugate (dilution 1:10,000, Agilent P026002-2) and anti-rabbit IgG HRP conjugate (dilution 1:10,000, Agilent P044801-2).

## Analysis of polymerase usage

For each sample approx. 200 million PE read were obtained. Raw reads were aligned to GRCh38 using Bowtie2 (version 2.3.5). Those which aligned to multiple genomic locations with the same mismatch scores (AS and XS scores as outputted by Bowtie2) were excluded using a custom Perl script: sam-dup-align-exclude-v2.pl (available at the GitHub site: https://github.com/yasukasu/sam-dup-align-exclude). The position of the 5′ end of each R1 read (which corresponds to the 5′ end of ssDNA hydrolysed by alkali treatment) was determined, and the number of reads in 1-kb bins across the genome were counted separately for the Watson and Crick strands using a custom Perl script: pe-sam-to-bincount.pl (available at the GitHub site: https://github.com/yasukasu/sam-to-bincount). This generated the four datasets in seprates csv files for the analysis of one polymerase.

In case of Polε: at the chromosome coordinate $x$, $N_w^\varepsilon(x)$, is the count for RNASEH2-mAID POLE1-M630F on the Watson strand; $N_c^\varepsilon(x)$ is the count for RNASEH2-mAID POLE1-M630F on the Crick strand; $N_w^+(x)$ is the count for RNASEH2-mAID POL$^+$ on the Watson strand; $N_c^+(x)$ is the count for RNASEH2-mAID POL$^+$ on the Crick strand. After obtaining the genome-wide read count data, genomic bins where the count of the polymerase mutant or the control cell line are less than 5 in both strands (e.g. $N_w^\varepsilon(x){<}5$ and $N_c^\varepsilon(x){<}5$) are excluded from further calculation for polymerase usage. The datasets were normalised using the total number of reads: e.g. $N'^\varepsilon_w(x) = N_w^\varepsilon(x)/\sum N_w^\varepsilon$ for the Polε mutant on the Watson strand (where N′ indicates normalisation). These normalised genomic bin data of Polε mutant were divided by those of the control cell line (polymerase proficient cells) to calculate relative polymerase usage: e.g. $E_w(x) = N'^\varepsilon_w(x)/N'^+_w(x)$ for usage of Polε on the Watson strand; $E_c(x) = N'^\varepsilon_c(x)/N'^+_c(x)$ for usage of Polε on the Crick strand. The equivalent analysis was performed to yield usage of Polα on Watson and Crick strand: $A_w(x)$ and $A_c(x)$. When these data were plotted or used for further analysis (below), they were smoothed using moving average of $2m+1$, where m is an arbitrary number the value of which is given in the relevant context. Thus, the data point for each bin is an average of $2m+1$ bins: the point of origin and the m bins either side. This analysis was performed using R-script: bincount-csv_to_pol-usage-wig.R (see Code availability).

## Initiation index and fork index

The difference between each neighbouring data point of polymerase usage was calculated as $\Delta E_w(x)$, $\Delta E_c(x)$, $\Delta A_w(x)$ and $\Delta A_c(x)$ with $E_w(x)$, $E_c(x)$, $A_w(x)$ and $A_c(x)$ which were smoothed using the value m = 30 (genome-wide plot, Figs. 2c, 3b, 5b, 6c, 7b, c) or m = 7 (the plot of averages or heat map, Figs. 3c–e, 5d–f). These differential data were

further smoothed by application of a moving average with value $m = 15$ (genome-wide plot) or $m = 7$ (the plot of averages or heat map). At each location where all four polymerase profiles exhibit consistent patterns for initiating bidirectional replication forks ($\Delta E_w(x){>}0 \cap \Delta E_c(x){<}0 \cap \Delta A_w(x){<}0 \cap \Delta A_c(x){>}0$) or patterns consistent with the merging of two forks ($\Delta E_w(x){<}0 \cap \Delta E_c(x){>}0 \cap \Delta A_w(x){>}0 \cap \Delta A_c(x){<}0$) an initiation index was defined as: $Ini(x) = \Delta E_w(x) - \Delta E_c(x) - \Delta A_w(x) + \Delta A_c(x)$. This data was subjected to $Z$-score normalisation (mean = 0, standard deviation = 1) and $Z(0)$ were subtracted to maintain the original + or − information, which represent increased levels of replication initiation and termination in the cell population respectively. This analysis was performed using R-script: pol-usage-wig_to_ini-index-wig.R (see Code availability).

Fork index was similarly calculated, but separately for rightward and leftward moving forks ($Fk^R$ and $Fk^L$). For example, locations of rightward fork initiation are where $\Delta E_w(x){>}0 \cap \Delta A_c(x){>}0$. Similarly, positions of rightward moving fork termination are where $\Delta E_w(x){<}0 \cap \Delta A_c(x){<}0$. At these positions Fk was calculated as $Fk^R(x) = \Delta E_w(x) + \Delta A_c(x)$ to give the fork index of rightward moving forks. The equivalent was performed for leftward moving forks: location initiation is where $\Delta E_c(x){<}0 \cap \Delta A_w(x){<}0$ and termination where ($\Delta E_c(x){>}0 \cap \Delta A_w(x){>}0$). The index for these locations was calculated as $Fk^L(x) = - dE_c(x) - dA_w(x)$. These data were subjected to $Z$-scores normalisation and $Z(0)$ subtracted. This analysis was performed using R-script: pol-usage-wig_to_fork-index-wig.R (see Code availability).

## RFDs from Pu-seq and Ok-seq

RFD from polymerase usage were calculated by subtraction of polymerase profiles typical of leftward moving fork signals from rightward moving fork signals[20]. When using only Polε usage data, $RFD^\varepsilon = (E_w(x) - E_c(x))/(E_w(x) + E_c(x))$. When using only Polα usage data, $RFD^\alpha = (- A_w(x) + A_c(x))/(A_w(x) + A_c(x))$. When using data from both polymerases, $RFD^{\varepsilon|\alpha} = (E_w(x) - E_c(x) - A_w(x) + A_c(x))/(E_w(x) + E_c(x) + A_w(x) + A_c(x))$. To calculate RFD from Ok-seq data, raw sequence read data from OK-seq experiments were obtained from the NCBI Sequence Read Archive (RPE1[21]: SRX4036932, GM06990[20]: SRX1427548, HeLa[20]: SRX1424656) and mapped and counted using the same pipelines as used for the Pu-seq data. Okazaki fragment counts on Watson and Crick strands were defined as $OK^W(x)$ and $OK^C(x)$ and the RFD was calculated as: $RFD^{OK} = (OK^C(x) - OK^W(x))/(OK^C(x) + OK^W(x))$[20]. The RFD datasets from Pu-seq and OK-seq were further smoothed by application of a moving average where $m = 3$. To convert $RFD^{\varepsilon|\alpha}$ to rightward fork proportion, the range between values at −3 standard deviation and +3 standard deviation, which covers 99.7% of data, were converted to 0 to 100%.

## Coupling index

The data of polymerase usage: $E_w(x)$, $E_c(x)$, $A_w(x)$ and $A_c(x)$ were smoothed using the value $m = 30$ (genome-wide or 2D plot, Figs. 6c, d, 7a–c) or $m = 7$ (the plot of averages or heat map, Fig. 6e, f). To establish a separate coupling index (CI) for both leftward and rightward forks, the lagging strand profile was subtracted from the leading strand profile. For rightward moving forks: $CI^R = (E_w(x) - A_c(x))/(E_w(x) + A_c(x))$, for leftward moving forks: $CI^L = (E_c(x) - A_w(x))/(E_c(x) + A_w(x))$.. This analysis was performed using R-script: pol-usage-wig_to_coupling-index-wig.R (see Code availability).

## RNA-seq data analysis

Extraction of total RNA from HCT116 + TetOsTIR1 cells was performed by using Monarch® Total RNA Miniprep Kit (New England Biolabs, Inc, #T2010S). cDNA library construction using TruSeq Stranded Total RNA Library Prep Kit with Ribo-Zero Human (Illumina #RS-122-2201) and sequencing on an Illumina NovaSeq6000 platform was performed by Macrogen (Tokyo, Japan). Approx. 21 million pair-end reads were sequenced. Raw sequenced reads were aligned to GRCh38 using STAR (version 2.7.3a, https://github.com/alexdobin/STAR). Annotations of

transcript units in the human genome (version 94) were retrieved from the Ensembl Genome Brower (http://www.ensembl.org) and those which are categorised as transcript support level 1 or 2 and Esembl/Havana-merged transcripts (i.e. high probability of correct annotation) were chosen for inclusion in the alignment target set. To obtain data of fragments per Kilobase of exon per Million fragments mapped (FPKM), bam files of mapped reads was analysed using Cufflinks (http://cole-trapnell-lab.github.io/cufflinks/).

## Statistics and reproducibility
Three biological replicates were obtained for datasets of Polε and Polα and both were used for all the analysis in this study. Data from one replicate are presented in Figs. 1–7: data from the second and third replicates showed excellent agreement and where relevant is shown in supplementary data. Data analysis and graphic visualisation was performed using R language.

## Reporting summary
Further information on research design is available in the Nature Portfolio Reporting Summary linked to this article.

## Data availability
The data that support this study are available from the corresponding author upon request. The Pu-seq data generated in this study, including raw sequencing reads and processed data for graphs in Figs. 2–7, have been deposited in the NCBI GEO under accession code GSE189668. High resolution RT data in HCT116 are available from GSE137764. OK-seq sequencing read data used in this study are available from NCBI SRA (SRP065949 and SRP144505). Data for genome-wide G4-duplex formation are available from GSE110582. ChIP-seq data for histone modification in HCT116 are available from GSE58638. End-seq data in HCT116 are available from GSE116321. The source data underlying Fig. 1b–f and Supplementary Fig. 1a–e, as well as uncropped blots are provided as Source Data files. Source data are provided with this paper.

## Code availability
The genome-wide profiles of polymerase usage, initiation index, fork index and coupling index were created using R. The codes are available at the author's GitHub site (https://github.com/yasukasu/Human_Pu-seq)[59].

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

## Acknowledgements

This work was supported by: JST PRESTO grant and FOREST program (JPMJPR18K7 and JPMJFR204X) to YD; JSPS KAKENHI grants (JP16H06151, JP17K19336, JP20H03233 and JP21K19203) to YD; (JP20H05396, JP21H04719 and JP22H04703) to MTK; Naito foundation, Takeda Science Foundation, Astellas foundation for research on metabolic disorders and NIG Collaborative Research Program grants (3B2016, 3A2017, 83A2019 and 81A2021) to YD; Wellcome Trust grant (110047/Z/15/Z) to AMC.

## Author contributions

Y.D. conceived the study. Y.D. M.T.K. and A.M.C. designed the experimental approaches. Y.D designed informatics approaches. E.K., Y.K., T.M, F.Y., T.N., A.H., T.O. and Y.D. performed experiments and their analysis. Y.D. performed computational analysis. Y.D wrote the manuscript. A.M.C, M.T.K. and T.O. edited the manuscript.

## Competing interests

The authors declare no competing interests.
