## [Peer Review File · Nature Communications]

Global landscape of replicative DNA polymerase usage in the human genomeREVIEWER COMMENTS

Reviewer #1 (Remarks to the Author):

In this manuscript, Koyanagi et al use HCT116 engineered to have mutations in DNA polymerases together with inducible degradation of RNASEH2A to perform Pu-seq in mammalian cells. They confirm the division of labor between DNA polymerase alpha and epsilon to the lagging and leading strand, respectively, in human cells. They then use PU-seq data to derive RFD profiles, which they show are similar to OK-seq RFD profiles. They then identify initiation sites or zones, and show that many of them have sizes of several tens of Kbs, and - notably - that efficient initiation can also be detected later in S phase. In some heterochromatic regions, replication appeared to be random. they then are able to compare replication activity with transcription activity and directionality and derive interesting results regarding fork slowing and termination as a consequence of head-on collisions. Last, they show interesting evidence of polymerase uncoupling at certain genomic regions, most notably at several fragile sites. Overall, this paper applies a very elegant technique to human cells and present several interesting results. The analyses are solid and the presentation is clear and engaging. I find this to be a useful and compelling paper.

- A notable weakness is the use of only HCT116 cells, which are applicable to this problem but represent a single cancer cell line, and a hypermutable one to begin with. This should be acknowledged for the least.
- Based on Figure 2, the data appears to be somewhat noisy in small scale. Thus, I would be cautious in interpreting the exact locations and width of initiation zones. Similarly, for replication initiation sites, I am not convinced that these represent true origin locations (peak positions within initiation zones) as opposed to being a consequence of the data and processing. In that respect, two experimental repetitions might not be sufficient. If the authors want to make a convincing statement based on this, they should first establish the statistical rigor of these designations. Along these lines, figure 3b compares their data to high-res repli-seq, but a more formal statistical analysis would be useful (I'm not even sure they necessarily need to use "high-resolution" repli-seq if it's not representable and analyzable as a single profile?).
- Similar to the above, a more formal analysis of how their initiation sites compare to those identified by other techniques would have been useful. Is Pu-seq more/less/equally accurate as some of those other techniques? Can this be shown statistically in a clear manner?
- The discussion mostly repeats results sections. I would suggest shortening the discussion or providing a more in-depth perspective of some of the results.
- In pages 4 and 5, they mention "control cell lines" several times. Please detail what these control cell lines are.
- Figure S2d- replace "chromosome VIII" with "chromosome 8"

Reviewer #2 (Remarks to the Author):

In this study, Koyanagi et al. investigate the usage of leading and lagging strand DNA polymerase epsilon and alpha across the human genome of HCT116 cells. They utilize the Pu-seq (polymerase usage) approach to track the incorporation of ribonucleotides across the human genome of HCT116 cells in which they introduced specific mutations of replicative DNA Pol ϵ and Pol α , POLE1-M630F and POLA1-Y865F, respectively, which incorporate an elevated level of ribonucleotides in DNA during DNA synthesis. The authors also employed a degron approach to minimize the activity of RNase H2 on ribonucleotides in DNA. By looking at the ratio of rNMP presence in the Pol ϵ -mutant vs Pol α -mutant cells the authors determine the polymerase usage across the genome. The study is like the one conducted for yeast DNA by Zhou et al. 2019 (PMID: 31495565), except that all analyses with DNA Pol delta are missing in the manuscript by Koyanagi et al. The study confirms that replication initiation is influenced by the transcriptional activity of the cells, and identifies replication initiation zones. This

is in line with previous studies about DNA replication in human cells. Moreover, it is found that the chromatin status is also important for shaping transcription initiation in the HCT116 cells, that there are regions of polymerase uncoupling in large genes, and that there is inhibition of leading strand synthesis at two common fragile sites.

The manuscript represents an interesting work and valuable for the field. Weakness is that mutant polymerase alleles are used instead of wild-type cells, the analyses are done in cells devoid of ribonuclease H2, and only two biological replicates of the experiments are presented. Comments have been provided to strengthen the study.

Comments:

- In this study two biological replicates were obtained for datasets of Pol ϵ and Pol α . It is not clear how many libraries of ribonucleotides were prepared for Pol alpha and Pol epsilon mutant cell lines. Is it two libraries for Pol ϵ and two for Pol α ? The GSE accession of this project appears invalid currently. If there are only two repeats, statistical analysis is not possible. A concern is whether two biological repeats are sufficient to support the conclusions. Why not having three repeats?

- Figure 1a is confusing. It should be indicated what the orange lines mean versus black lines, and what the orange dots mean. The scheme as it is shown seems to suggest that the ribonucleotides (R) are captured by the Pu-seq procedure, while this is not the case. Also the dotted boxes suggest that DNA sequences on both sides of the breaks caused by alkali are sequenced, and this is not the case of Pu-seq. The figure should be modified and the legend also accordingly.

- The Pu-seq protocol, in addition to capturing sites of ribonucleotides in DNA by capturing the sequence immediately downstream of the ribonucleotides after alkali treatment, can capture sites of primers of Okazaki fragments with the same efficiency as ribonucleotides. Could this affect the accuracy of the results of this study?

- A major criticism of the previous yeast work that utilized low fidelity polymerase mutants incorporating more ribonucleotides as markers for the division of labor between leading and lagging strand polymerases was that most of the data had been obtained in mutant polymerase rather than in wild-type polymerase cells, as well as in mutant RNase H2 cells. Considering that RNase H2 function is not limited to cleavage at ribonucleotides embedded in DNA, but it also cleaves long RNA/DNA hybrids like those of primers in Okazaki fragments, I am wondering whether defect in RNase H2 may delay primer removal, alter the synthesis of the lagging strand and may even facilitate uncoupling between leading and lagging polymerases. How can RNase H2 defect impact the results?

- In this study, the degron AID system is used to deactivate RNase H2 subunit A. What are the pros and cons of this system compared to generating a null mutation in RNase H2A?

- There may be PCR duplicates generated during the Pu-seq procedure. How do the PCR duplicates affect the results of the study? Are there any steps performed to remove the PCR duplicates in this study?

- There are other studies that have identified replication initiation zones in human DNA. Compared to the initiation zones identifies by the other approaches, what is the advantage of the Pu-seq approach? It would be valuable if the authors could discuss which initiation zones are similar among the different studies and which are different and how could be differences be explained.

- Too many abbreviations make the paper a little hard to follow. RFD, FI, CI, CD, CV, RT.

- Page 7, from line 225, what are the differences between the genes with initiation zones at TSS and the genes without initiation zones? Are there any common features?

- Page 8, from line 252. Is it possible to separate the two signals of co-directional TSS (“...a negative signal (fork termination) at and immediately upstream of TSS is embedded within strong positive signals derived from initiation events associated with the upstream 20 kb”)? Do these signals have the same magnitude? Which genes have the initiation zones upstream of the TSS? Which genes have the termination zones at TSS?

- Page 8, from line 252, this paragraph is also hard to follow.

- Fig. 5d Color scale is not linear scale. The r value (Pearson’s r) is not high (> 0.5), and the picture looks like a positive correlation slope=1 with a negative one, slope = -1. Is it possible to separate them into two groups? Are there any potential relationship between the two groups?

- Page 11, line 363, is there any reason the Pol delta mutants in human cannot be isolated compared to yeast Pol δ , human Pol α and Pol ϵ ? In the Zhou et al. 2019 (PMID: 31488849), they compared pol ϵ with pol α + pol δ , in this paper pol α is only compared with pol α . Is there any bias if pol δ is not considered?

- Page 11, line 375, inter peak distance or intra peak distance?

- In Figure 1d, the labelling of the gel is shifted and the (kb) label is missing for the marker.

- Are data shown in Figure 1 coming just from one sample? Are the results reproducible? Can the smears be quantified?

Other comments on reference citations

- The statement “To expand the analysis of polymerase usage genome-wide, the locations of the increased levels of rNMPs incorporated by individual mutated DNA polymerases (Pol α , Pol δ or Pol ϵ) were identified by whole genome sequencing5-7.” should also include the reference by Koh et al 2015 (PMID: 25622106) reporting comparable results and published at the same time as the three other references cited.

- Also it would appropriate to include the citation of Xu and Storici, 2020 (PMID: 34551434) that analyzed data both from yeast mutant and wild type DNA Pols also in wild-type RNase H2 cells.

- Page 4, line 86. The statement “Ribonucleotides are normally incorporated by the replicative polymerases approximately 1:4000 incorporation events.” should be supported by a reference. Similarly, the next sentences lack references.

- Page 5, line 90. “Therefore, RNase H2 must be inactivated concomitantly with increased rNMP incorporation in order to map the distribution of rNMPs genome-wide.”
The statement does not reflect the literature. Mapping of ribonucleotides in DNA of wild-type RNase H2 cells have been for various species, e.g. work in different yeast species and in *C. reinhardtii* (PMID: 32415081 and PMID: 33490913).

Reviewer #3 (Remarks to the Author):

The authors (Koyanagi et al) describe for the first time polymerase usage sequencing (Pu-seq) in engineered human cells. Following RNaseH2 degradation in mutant DNA pol E and DNA pol A HCT116 cells (preferentially incorporating ribonucleotides), alkaline hydrolysis followed by DNA-sequencing library prep allows for multiple downstream methods of analysis to explore replication dynamics. The

authors find Pu-seq profiles correspond with previously published Okazaki-fragment (Ok-seq) sequencing and initiation sequencing (Ini-seq) data. Consistent with these reports, the authors find replication initiation is enriched at transcription regulatory units in a gene expression and length-dependent manner. The authors propose transcription machinery impacts replication fork progression in both codirectional and head-on orientations. Finally, the authors propose specific regions of the genome are enriched for replication strand synthesis decoupling, particularly at two common fragile sites, implicating the use of Pu-seq in studying mechanisms of genomic instability.

The Pu-seq method described in this report is a potentially exciting technique that adds to existing work mapping genome-wide replication fork dynamics in human cells. Some strengths of this technique are the relatively few starting cells needed, collection from asynchronous cells, and feasible workflow. Initiation and replication fork directionality profiles are consistent with and validate the current understanding of polymerase usage, replication timing, and replication initiation orientation around transcription. The authors highlight the potential of this technique through multiple analysis methods mapping unique measures of replication measure (initiation, termination, directionality, decoupling). Specifically, the authors report genome-wide analysis of strand decoupling in human cells for the first time, uniquely measurable by Pu-seq. This analysis demonstrates potential for future clinical correlations to genomic instability at common fragile sites and disease / cancer.

The authors are missing important controls for validating the use and robustness of this technique. These lack of controls make it difficult to support the authors' conclusions. Specifically, the authors' interpretation of mapping replication fork interaction with transcription machinery and strand decoupling needs to be further validated, as lack of thereof could mean the authors misinterpreted some major conclusions. This is a major weakness to the study and some of the results based on the uncoupling are quite difficult to reconcile, given that there are no biological precedent for uncoupling that gives more POL Epsilon (leading) synthesis (i.e. difficult to envision how lagging strand synthesis (POL delta and POL alpha) could be blocked given that it is already discontinuous synthesis that can easily just reprime downstream of a lesion). The authors also missed opportunities to present or interpret specific results (see below comments). A comprehensive graphical summary of the proposed mechanism linking transcription-replication conflicts to strand decoupling at convergent and codirectional collisions, as the authors propose, would be helpful. Overall, the long-distance range of uncoupling of the leading/lagging strand polymerases is quite difficult to envision mechanistically, and would require more experimental controls to support their provocative conclusions.

Major points:

1. The study lacks sufficient controls (such as western blot of checkpoint activation, cell cycle dynamics) to show that transient RH2 loss (24, 48, 72 hrs) is not indirectly affecting replication dynamics / cell growth and/or causing replication stress. This could be especially important to look at earlier time points of RH2 loss (12h, 24h), before the 48h time point used in Pu-seq analysis. Based on the subsequent results that mostly align with other ok-seq data sets, it might be workable, but the cells are dividing up to 2 cell cycles (48h time point) without RH2 prior to Pu-seq. This could alter origin usage and fork progression, consistent with the authors' citation that RH2 is essential for growth in this p53-wildtype cell line. It could also be helpful if this technique is exploited in future experimental conditions that might be more sensitive to RH2 loss. Nevertheless, the paradox of causing ribonucleotide incorporations as a way to mark strand-specific synthesis, yet the mis-incorporation of ribonucleotides itself causes fork stalling and mutagenesis and DNA damage is quite disconcerting. More controls are needed to show that the experimental conditions have to be just right to achieve a balance between not creating too much DNA damage to alter replication dynamics indirectly, yet sufficient incorporation to sequence and map replication sites.
2. What are the control lines used for Pu-seq analysis shown? WT polymerases with or without IAA induction? I think this should be stated in the main text.
3. What does the polymerase usage look like not normalized to control lines? In theory wouldn't isolation / alkaline treatment from control lines (no mutant and/or no IAA induction), result in little to

no gDNA? What's the rationale behind normalizing to a control line and not only to total read count?
Signal amplification?

4. Lines 225-230 / Extended Figure 6A: The authors conclude that ~20% of initiation zones don't map within regions ~30kb upstream of TSS of transcribed genes ("a-1" in Extended Figure 6). It could be interesting to see whether these 20% of genes are enriched for long genes, where persistent transcription machinery could be present. Comparing Main Figures 3C and 3D, it seems like the association of initiation is somewhat stronger at TSS and TTS of very long genes vs very those of very highly transcribed genes. If these 20% of genes are enriched for longer genes, the conclusion that there is a subset of "transcriptionally active genes not necessarily associated with replication initiation" is maybe incomplete.

5. Extended Figure 6B: I think this figure is confusing. The authors describe 60% of initiation zones in both replicates don't map to either 30kb upstream of a TSS or downstream of a TTS. How was this analysis done? Do these 60% of origins represent less efficient origins? If they're trying to show the majority of origins are intergenic, that could be shown easily. 30kb is also a large region to be looking within and likely encompasses ~50% of genes. Are some of these origins intragenic? I think furthering this analysis would help the authors' argument that a significant portion of origins don't map to transcriptional regulatory units, or maybe it could be left out.

6. Line 239 – 243 / Main Figure 4B: It would be helpful for ease of understanding to mention the difference between fork index and the previously described initiation index, that the fork indices are mutually exclusive except for regions of overlap which correspond to the initiation index.

7. Main figure 4D,E: To support the authors argument on the interpretation of the codirectional fork profile they should definitely show a subset of codirectional forks where there are no termination events immediately upstream of the TSS, if they argue the profile they see is a mixed signal of termination and initiation of multiple forks. It appears CD forks in general are associated with more termination than CV forks regardless of origin status at TSS (Extended Figure 6C, comparing "a-1" bottom and top). However, there still appears to be a relative termination within CD origins at TSSs (Extended Figure 6C, top, "a-2"). As it stands the data from extended figure 6C and figure 4D suggests that every single codirectional fork is associated with an upstream termination, which is a different mechanism than the authors propose in the text.

8. Main Figure 5: One important control would be to show that this is uncoupling and not due to differing efficiency in mutant polymerase activity possibly at specific regions of the genome, or alternate polymerase usage. Also, it is super surprising that uncoupling can last over 1 Mb in length. If this truly was the case, that is a lot of exposed ssDNA and checkpoint activation and single/double strand DNA breaks occurs within this region. Activation checkpoints would alter replication dynamics indirectly so it is difficult to know what is the "cause" and "effect" of these changes.

9. Line 349 / Figure 6B: It appears that the low CI does correlate with a rightward fork initiation and leftward fork termination, contrary to the author's argument.

10. A graphical figure incorporating a model of transcription-replication collisions at CD and CV forks, plus the distinct decoupling profiles at each of these TSSs could be helpful.

Minor points:

1. Main Figure 1B: Add clarifying text above respective blots indicated AID vs AID2 system

2. Line 109-110: I'm not sure this last sentence is necessary, since there is no data in this study on mutant POLD overexpression. Also, the point isn't clear : ("mutant POLD does not provide essential function"); through what assays was this conclusion made?

3. "Co-directional" is misspelled throughout the figures

4. It could be helpful to depict a termination event calculation similar to in figure 2B for ease of understanding how negative initiation values / fork indices correspond to termination events

5. Line 262: "The combined pattern of CV and CD forks is consistent with the trend for fork initiation in both orientations 20kb upstream of TSS...". It would be useful to plot the 0% CV and CD lines at TSS and TTS seen in 4D on a separate graph in the supplement to more easily understand the switch from left/right fork indexes in Main Figure 4B to convergent / codirectional in Figure 4D and onward. This will allow readers to see the leftward index corresponds to convergent forks and right fork index

corresponds to codirectional forks, and understand the mapped replication fork directionality without influence of transcription. This figure could be referenced after this clause on line 262-263.

6. Figure 4E: 0.75 label is negative

7. Extended Figure 8A: what do 0-50% and 50-100% refer to? Percentile of origin efficiency?

8. Main Figure 5D: label scale bar missing

9. Line 329 – 330: sentence typo

10. Figure 6B: grey boxes interrupting view of zoomed in

11. Line 432 – 433: missing citation ([https://www.cell.com/cell-reports/pdf/S2211-1247\(21\)00072-3.pdf](https://www.cell.com/cell-reports/pdf/S2211-1247(21)00072-3.pdf))

RESPONSE TO REVIEWERS' COMMENTS

We thank all referees for their thoughtful and constructive comments. Below we detail our response to each comment (in bold text) and highlight changes made to the manuscript in response to these.

Reviewer #1 (Remarks to the Author):

In this manuscript, Koyanagi et al use HCT116 engineered to have mutations in DNA polymerases together with inducible degradation of RNASEH2A to perform Pu-seq in mammalian cells. They confirm the division of labor between DNA polymerase alpha and epsilon to the lagging and leading strand, respectively, in human cells. They then use PU-seq data to derive RFD profiles, which they show are similar to OK-seq RFD profiles. They then identify initiation sites or zones, and show that many of them have sizes of several tens of Kbs, and - notably - that efficient initiation can also be detected later in S phase. In some heterochromatic regions, replication appeared to be random. they then are able to compare replication activity with transcription activity and directionality and derive interesting results regarding fork slowing and termination as a consequence of head-on collisions. Last, they show interesting evidence of polymerase uncoupling at certain genomic regions, most notably at several fragile sites. Overall, this paper applies a very elegant technique to human cells and present several interesting results. The analyses are solid and the presentation is clear and engaging. I find this to be a useful and compelling paper.

- A notable weakness is the use of only HCT116 cells, which are applicable to this problem but represent a single cancer cell line, and a hypermutable one to begin with. This should be acknowledged for the least.

As suggested, we added the following sentence in the last paragraph of discussion.

“Although we have currently only applied Pu-seq to a single metazoan cell line, HCT116, which is derived from colon cancer and hypermutable, we anticipate that by constructing the system in different cell lines harbouring distinct developmental or genetic backgrounds the methodology will allow detailed analysis of the intrinsic flexibility of DNA replication.”

- Based on Figure 2, the data appears to be somewhat noisy in small scale. Thus, I would be cautious in interpreting the exact locations and width of initiation zones. Similarly, for replication initiation sites, I am not convinced that these represent true origin locations (peak positions within initiation zones) as opposed to being a consequence of the data and processing. In that respect, two experimental repetitions might not be sufficient. If the authors want to make a convincing statement based on this, they should first establish the statistical rigor of these designations.

In term of peaks within initiation zone, we apologise for not making our thoughts clear. We do not mean that these are ‘replication origins’ at which initiation exclusively occurs. Rather, we isolated these as the reference location to examine the correlation of various genomic features with activity of replication initiation (e.g. Supplementary Fig. 9 in the revised version). We modified the context which may have caused the wrong impression. We also changed the section title ‘Local genomic features at replication initiation site’ to ‘~ within initiation zones’ and text of this section.

In term of analysis on initiation zones, we performed analysis by adding one more experimental replicate of Pu-seq (Supplementary Fig. 6cd). This result confirmed that the efficient initiation zones are reproductively detected in all three replicates and their average sizes are also very similar, while we also further clarify the size of initiation zone is highly diverse (in the section ‘Defining replication initiation regions from polymerase usage’).

Along these lines, figure 3b compares their data to high-res repli-seq, but a more formal statistical analysis would be useful (I’m not even sure they necessarily need to use “high-resolution” repli-seq if it’s not representable and analyzable as a single profile?).

Unfortunately, we could not obtain data of initiation zone derived from repli-seq in HCT116 from the GEO depository (GSE137764) of the original paper (although the one in ES cells is deposited). It is hard for us to devise the new method to analyse correlation of replication timing (RT) profiles and initiation index. However, we believe that visual inspection of these two profiles together show colocalization of peaks of RT and initiation index and confirm the consistency of data derived from repli-seq and Pu-seq (please also check supplementary Fig. 7a). For this reason, we wish to keep the comparison of RT and initiation index (Fig.3b and Supplementary Fig. 7a)

- Similar to the above, a more formal analysis of how their initiation sites compare to those identified by other techniques would have been useful. Is Pu-seq more/less/equally accurate as some of those other techniques? Can this be shown statistically in a clear manner?

We added a pairwise comparison with data derived from OK-seq, SNS-seq, bubble-seq and ini-seq (Supplementary Fig. 7b) and described our interpretation in 3rd paragraph of the section ‘Defining replication initiation regions from polymerase usage’.

- The discussion mostly repeats results sections. I would suggest shortening the discussion or providing a more in-depth perspective of some of the results.

As suggested by the reviewer, we removed some repeats and shortened the context in the section of discussion.

- In pages 4 and 5, they mention “control cell lines” several times. Please detail what these control cell lines are.

We induced RNASEH2A degradation in both the polymerase mutant and wild-type polymerase cell lines. The data derived from wild-type polymerase cell line in this condition are used as the control. We clarified this point in the main text (please see the first paragraph of the section ‘Mapping polymerase usage across the genome’).

- Figure S2d- replace “chromosome VIII” with “chromosome 8”

Corrected

Reviewer #2 (Remarks to the Author):

In this study, Koyanagi et al. investigate the usage of leading and lagging strand DNA polymerase epsilon and alpha across the human genome of HCT116 cells. They utilize the Pu-seq (polymerase usage) approach to track the incorporation of ribonucleotides across the human genome of HCT116 cells in which they introduced specific mutations of replicative DNA Pol ϵ and Pol α , POLE1-M630F and POLA1-Y865F, respectively, which incorporate an elevated level of ribonucleotides in DNA during DNA synthesis. The authors also employed a degnon approach to minimize the activity of RNase H2 on ribonucleotides in DNA. By looking at the ratio of rNMP presence in the Pol ϵ -mutant vs Pol α -mutant cells the authors determine the polymerase usage across the genome. The study is like the one conducted for yeast DNA by Zhou et al. 2019 (PMID: 31495565), except that all analyses with DNA Pol delta are missing in the manuscript by Koyanagi et al. The study confirms that replication initiation is influenced by the transcriptional activity of the cells, and identifies replication initiation zones. This is in line with previous studies about DNA replication in human cells. Moreover, it is found that the chromatin status is also important for shaping transcription initiation in the HCT116 cells, that there are regions of polymerase uncoupling in large genes, and that there is inhibition of leading strand synthesis at two common fragile sites.

The manuscript represents an interesting work and valuable for the field. Weakness is that mutant polymerase alleles are used instead of wild-type cells, the analyses are done in cells devoid of ribonuclease H2, and only two biological replicates of the experiments are presented. Comments have been provided to strengthen the study.

Comments:

- In this study two biological replicates were obtained for datasets of Pol ϵ and Pol α . It is not clear how many libraries of ribonucleotides were prepared for Pol alpha and Pol epsilon mutant cell lines. Is it two libraries for Pol ϵ and two for Pol α ? The GSE accession of this project appears invalid currently. If there are only two repeats, statistical analysis is not possible. A concern is whether two biological repeats are sufficient to support the conclusions. Why not having three repeats?

To generate two experimental repeats of one polymerase profile, we independently made two libraries from the polymerase mutant (POLE1-M630F or POLA1-Y865F) and two libraries independently from the wild type polymerase cell lines. In the revised manuscript, we added one more replicate and data from multiple replicates were updated (Supplementary Fig. 3, Supplementary Fig. 6cd, Supplementary Fig. 7b, Supplementary Fig. 11b, Supplementary Fig. 8c, etc. and data of initiation zone described in the text). We also provide the access link to the data deposition site (NCBI GEO) and its security token.

<https://www.ncbi.nlm.nih.gov/geo/query/acc.cgi?acc=GSE189668>

The security token: odwzwyumlzmpstj

- Figure 1a is confusing. It should be indicated what the orange lines mean versus black lines, and what the orange dots mean. The scheme as it is shown seems to suggest that the ribonucleotides (R) are captured by the Pu-seq procedure, while this is not the case. Also the dotted boxes suggest that DNA sequences on both sides of the breaks caused by alkali are sequenced, and this is not the case of Pu-seq. The figure should be modified and the legend also accordingly.

We apology for not providing sufficient information for this figure. We add the explanation of ochre (or orange) lines in the figure legend. In Pu-seq, we are actually able to locate ribonucleotides in a single nucleotide resolution by mapping the 5' end of alkaline-cleaved DNA fragments. The dotted boxes presented to indicate the isolation of small ssDNA fragment by gel extraction. However, the position of dotted box was not accurate and thus we have modified this. These points are clarified in the updated Fig.1A and its legend.

- The Pu-seq protocol, in addition to capturing sites of ribonucleotides in DNA by capturing the sequence immediately downstream of the ribonucleotides after alkali treatment, can capture sites of primers of Okazaki fragments with the same efficiency as ribonucleotides. Could this affect the accuracy of the results of this study?

We understand this concern and are confident that this is not the case. We have added the mapping data derived from the control cell line; Pol⁺ cells in which RNASEH2A is degraded, to Supplementary Fig. 2 to highlight this. In these data, we do not observed similarity to the

pattern of lagging strand (or Pol α) profile, indicating that a detectable amount of Okazaki fragments were not extracted with alkaline-cleaved ssDNA. Presumably, without additional procedure (e.g. inactivation of DNA ligase (Smith et al Nature 483 434–438 2012) or purification of nascent strands (Petryk et al Nat Commun 7 10208 2016)), we do not extract significant Okazaki fragments from genomic DNA or identify their RNA primers as they are short-lived.

- A major criticism of the previous yeast work that utilized low fidelity polymerase mutants incorporating more ribonucleotides as markers for the division of labor between leading and lagging strand polymerases was that most of the data had been obtained in mutant polymerase rather than in wild-type polymerase cells, as well as in mutant RNase H2 cells. Considering that RNase H2 function is not limited to cleavage at ribonucleotides embedded in DNA, but it also cleaves long RNA/DNA hybrids like those of primers in Okazaki fragments, I am wondering whether defect in RNase H2 may delay primer removal, alter the synthesis of the lagging strand and may even facilitate uncoupling between leading and lagging polymerases. How can RNase H2 defect impact the results?

In a new Supplementary Fig. 1, we provide the evidence that degradation of RNASEH2A does not cause apparent effects in cell growth or the rate of DNA synthesis. We also monitor the levels of RNA/DNA hybrids (or RNASEH sensitive structures detected by S9.6 antibody) by a dot blot assay. This shows that RNA/DNA hybrids are not increased following degradation of RNASEH2A (at least by the time point of sampling; 48hr, Supplementary Fig. 1d). Collectively, these results indicated that the degradation of RNASEH2A is unlikely to influence overall kinetics of genome replication in the experimental conditions used in this study. Furthermore, our control cells where RNASEH2A is degraded in a Pol $^+$ background show no evidence of RNASEH2 affecting Okazaki fragment primers. However, we cannot fully exclude the possibility of a local effect of RNase H2 degradation at a specific genomic location. Thus, we add the following sentence in the first paragraph of discussion to clarify the pros and cons of ribonucleotide mapping technique.

'Our approach requires transient degradation of RNASEH2A, which could potentially perturb replication dynamics by causing replication stress. However, during the 48 hours in which we inactivate RNASHA, Chk1 is not phosphorylated, BrdU incorporation is not perturbed and we see no global increase in DNA/RNA hybrid structures recognised by the S9.6 antibody. We thus conclude that RNASH2A degradation does not influence the global trend of DNA replication.'

- In this study, the degron AID system is used to deactivate RNase H2 subunit A. What are the pros and cons of this system compared to generating a null mutation in RNase H2A?

As mentioned in the text (the first paragraph of ‘Construction of ribonucleotide-incorporating DNA polymerase mutant lines’), the function of RNASEH2A is essential for growth of p53 positive mammalian cells and thus we chose to inactivate RNASEH2 transiently. We reasoned that, compared to a KO, transient inactivation would allow sufficient labelling with rNMP but minimise chronic DNA damage and avoid any clonal or accumulative effects that may appear while maintaining KO cell lines.

- There may be PCR duplicates generated during the Pu-seq procedure. How do the PCR duplicates affect the results of the study? Are there any steps performed to remove the PCR duplicates in this study?

We did not perform the removal of duplicated sequence reads after mapping of the reads. We think this process is not always beneficial – PE reads of the same sequence are not always derived from PCR and their source may have already existed in the sample before PCR amplification. In Pu-seq, we recover at least 50 ng of ssDNA from the control cell lines and 400-600 ng from polymerase mutants (measured by OliGreen ssDNA Assay Kit; O7582). This quantity is generally far more than is typical for some ChIP-seq experiments or other HTS methods which start from small quantities of DNA. Thus, our amplification by PCR is relatively mild (ten cycle) and the ratio of PCR-duplicated reads will be very limited in the outputted sequenced data.

- There are other studies that have identified replication initiation zones in human DNA. Compared to the initiation zones identifies by the other approaches, what is the advantage of the Pu-seq approach? It would be valuable if the authors could discuss which initiation zones are similar among the different studies and which are different and how could be differences be explained.

We added a pairwise comparison with data derived from OK-seq, SNS-seq, bubble-seq and ini-seq (Supplementary Fig. 7b) and described our interpretation in 3rd paragraph of the section ‘Defining replication initiation regions from polymerase usage’.

- Too many abbreviations make the paper a little hard to follow. RFD, FI, CI, CD, CV, RT.

In the revised version, CI and FI are converted to original terms; ‘coupling index’ and ‘fork index’ in the results/discussion sections. However, we keep these abbreviations in formulas in materials and methods and figures. On the other hand, as RFD (replication fork directionality) and RT (replication timing) are generally used in other publications, we keep these abbreviations in the text.

- Page 7, from line 225 (**Page 8 line 255 in revised manuscript**), what are the differences between the genes with initiation zones at TSS and the genes without initiation zones? Are there any common features?

We tried additional analysis to investigate this point by incorporating various genomics features (ChIP-seq data of transcriptional factors or chromosome components, etc) or gene ontology analysis, etc. However, currently we are not able provide any striking evidence to specify the differences between the genes with initiation zones at TSS and the genes without initiation zones. Presumably, we required further large scale of analysis to identify how initiation zone are specially located in the nuclear structure. Thus, we wish to include this issue as a consideration for future studies and described this point in discussion.

- Page 8, from line 252 (**Page 9 line 291 in revised manuscript**). Is it possible to separate the two signals of co-directional TSS (“...a negative signal (fork termination) at and immediately upstream of TSS is embedded within strong positive signals derived from initiation events associated with the upstream 20 kb”)? Do these signals have the same magnitude? Which genes have the initiation zones upstream of the TSS? Which genes have the termination zones at TSS?

We added a heatmap which shows the trends of fork index in the individual regions upstream of active TSS (Fig. 5e, Supplementary Fig. 8c). This data clearly shows the heterogenous patterns of fork index – in the case of co-directional forks in these upstream regions, the signal of termination (blue) and initiation (ochre) are mixed and these numbers are nearly equal (initiation: 55.6%, termination’43.9%, Fig. 5e). On the other hand, the pattern of CV fork also contains both initiation and termination but regions with initiation are outnumbered (75.7%).

So far, we could not address what discriminates genes with termination and initiation around TSS. We probably need further large scale of analysis together with RNA polymerase dynamics and, local nuclear environment, etc. we wish to include this issue as a consideration for future studies.

- Page 8, from line 252 (**Page 9 line 291 in revised manuscript**), this paragraph is also hard to follow.

We simplified this sentence.

‘...two low positive peaks are evident. We interpret this as a combination of initiation and

termination: i.e. a negative signal (fork termination) at 0-20 kb upstream of TSS is embedded within a strong positive signal (fork initiation)'

- Fig. 5d (**Fig.6d in the revised version**), Color scale is not linear scale. The r value (Pearson's r) is not high (> 0.5), and the picture looks like a positive correlation slope=1 with a negative one, slope = -1. Is it possible to separate them into two groups? Are there any potential relationship between the two groups?

To visualise the sparse area within the distribution, we intentionally use the non-linear but exponential colour scale. As we show in the separate plots of chromosomes (Chr. 3 and Chr. 6 in Fig 7a in the revised version), there are outliers of which both rightward and leftward fork indices are critically low. As this subpopulation (bottom right area in Fig.6d) is observed only in a fraction of chromosomes, we do not consider this as the part of the negative (slope = -1) global trend.

- Page 11, line 363 (**Page 12 line 413 in the revised manuscript**), is there any reason the Pol delta mutants in human cannot be isolated compared to yeast Pol δ , human Pol α and Pol ϵ ?

We could answer this exactly: POLD1- L606G and L606M are lethal in the cell line. We speculate that, since the structure of human Pol δ is not identical to those of the yeasts, these mutations likely influence the polymerase activity itself.

In the Zhou et al. 2019 (PMID: 31488849), they compared pol ϵ with pol α + pol δ , in this paper pol α is only compared with pol α . Is there any bias if pol δ is not considered?

In canonical replication forks, all lagging strands are extended by Pol α after RNA priming and then Pol α subsequently is replaced by Pol δ for further synthesis. Thus, being based on this mechanism, the genomic distributions of Pol α and Pol δ are predicted to be very similar. Indeed, the profile of Pol α is not distinguishable from lagging strand profile of Pol δ as reported in *S. cerevisiae* (Clausen et al 2015, Reijns et al 2015). A similar observation has recently been reported for *S. pombe* (Naiman et al, 2021). To our knowledge, Zhou et al aimed to quantify the ratio of Pol δ synthesis compared to overall synthesis and therefore they needed to calculate the lagging strand profile from Pol α + Pol δ . Of course, in ideal conditions we would have been able to create and analyse a Pol δ mutant. However, we feel confident that the limitations of not having this data do not affect the conclusions that we can draw from the ϵ/α analysis.

- Page 11, line 375 (**Page 13 line 423 in the revised manuscript**), inter peak distance or intra

peak distance?

We apologised for this mistake, 'inter peak distance' is correct. Corrected.

- In Figure 1d, the labelling of the gel is shifted and the (kb) label is missing for the marker.

Thank you for pointing this out. We corrected this figure.

- Are data shown in Figure 1 coming just from one sample? Are the results reproducible? Can the smears be quantified?

We added the same data from replicate 2 and 3 as well as quantification the ratio of fragmented DNA (<2kb), which is the size of DNA used for the library preparation (Supplementary Fig. 1a).

Other comments on reference citations

- The statement "To expand the analysis of polymerase usage genome-wide, the locations of the increased levels of rNMPs incorporated by individual mutated DNA polymerases (Pol α , Pol δ or Pol ϵ) were identified by whole genome sequencing⁵⁻⁷." should also include the reference by Koh et al 2015 (PMID: 25622106) reporting comparable results and published at the same time as the three other references cited.

We apologise for missing the citation – we added Koh et al 2015 into the reference list together with citations 5-7.

- Also it would appropriate to include the citation of Xu and Storici, 2020 (PMID: 34551434) that analyzed data both from yeast mutant and wild type DNA Pols also in wild-type RNase H2 cells.

Thank you for guiding us to this recent publication – we cited this study as analysis of polymerase usage using the signature of rNMP incorporation by wild-type polymerases.

- Page 4, line 86. The statement "Ribonucleotides are normally incorporated by the replicative polymerases approximately 1:4000 incorporation events." should be supported by a reference. Similarly, the next sentences lack references.

We cited the recent review article from Williams and Kunkel (2022) and accordingly modified

the number of rNMP incorporation.

- Page 5, line 90. "Therefore, RNase H2 must be inactivated concomitantly with increased rNMP incorporation in order to map the distribution of rNMPs genome-wide."

The statement does not reflect the literature. Mapping of ribonucleotides in DNA of wild-type RNase H2 cells have been for various species, e.g. work in different yeast species and in *C. reinhardtii* (PMID: 32415081 and PMID: 33490913).

Thank you for pointing out – we removed this sentence from the main text.

Reviewer #3 (Remarks to the Author):

The authors (Koyanagi et al) describe for the first time polymerase usage sequencing (Pu-seq) in engineered human cells. Following RNaseH2 degradation in mutant DNA pol E and DNA pol A HCT116 cells (preferentially incorporating ribonucleotides), alkaline hydrolysis followed by DNA-sequencing library prep allows for multiple downstream methods of analysis to explore replication dynamics. The authors find Pu-seq profiles correspond with previously published Okazaki-fragment (Ok-seq) sequencing and initiation sequencing (Ini-seq) data. Consistent with these reports, the authors find replication initiation is enriched at transcription regulatory units in a gene expression and length-dependent manner. The authors propose transcription machinery impacts replication fork progression in both codirectional and head-on orientations. Finally, the authors propose specific regions of the genome are enriched for replication strand synthesis decoupling, particularly at two common fragile sites, implicating the use of Pu-seq in studying mechanisms of genomic instability.

The Pu-seq method described in this report is a potentially exciting technique that adds to existing work mapping genome-wide replication fork dynamics in human cells. Some strengths of this technique are the relatively few starting cells needed, collection from asynchronous cells, and feasible workflow. Initiation and replication fork directionality profiles are consistent with and validate the current understanding of polymerase usage, replication timing, and replication initiation orientation around transcription. The authors highlight the potential of this technique through multiple analysis methods mapping unique measures of replication measure (initiation, termination, directionality, decoupling). Specifically, the authors report genome-wide analysis of strand decoupling in human cells for the first time, uniquely measurable by Pu-seq. This analysis demonstrates potential for future clinical correlations to genomic instability at common fragile sites and disease / cancer.

The authors are missing important controls for validating the use and robustness of this technique. These lack of controls make it difficult to support the authors' conclusions. Specifically, the authors' interpretation of mapping replication fork interaction with transcription machinery and strand decoupling needs to be further validated, as lack of thereof could mean the authors misinterpreted some major conclusions. This is a major weakness to the study and some of the results based on the uncoupling are quite difficult to reconcile, given that there are no biological precedent for uncoupling that gives more POL Epsilon (leading) synthesis (i.e. difficult to envision how lagging strand synthesis (POL delta and POL alpha) could be blocked given that it is already discontinuous synthesis that can easily just reprime downstream of a lesion). The authors also missed opportunities to present or interpret specific results (see below comments). A comprehensive graphical summary of the proposed mechanism linking transcription-replication conflicts to strand decoupling at convergent and codirectional collisions, as the authors propose, would be helpful. Overall, the long-distance range of uncoupling of the leading/lagging strand polymerases is quite difficult to envision mechanistically, and would require more experimental controls to support their provocative conclusions.

Major points:

1. The study lacks sufficient controls (such as western blot of checkpoint activation, cell cycle dynamics) to show that transient RH2 loss (24, 48, 72 hrs) is not indirectly affecting replication dynamics / cell growth and/or causing replication stress. This could be especially important to look at earlier time points of RH2 loss (12h, 24h), before the 48h time point used in Pu-seq analysis. Based on the subsequent results that mostly align with other ok-seq data sets, it might be workable, but the cells are dividing up to 2 cell cycles (48h time point) without RH2 prior to Pu-seq. This could alter origin usage and fork progression, consistent with the authors' citation that RH2 is essential for growth in this p53-wildtype cell line. It could also be helpful if this technique is exploited in future experimental conditions that might be more sensitive to RH2 loss. Nevertheless, the paradox of causing ribonucleotide incorporations as a way to mark strand-specific synthesis, yet the mis-incorporation of ribonucleotides itself causes fork stalling and mutagenesis and DNA damage is quite disconcerting. More controls are needed to show that the experimental conditions have to be just right to achieve a balance between not creating too much DNA damage to alter replication dynamics indirectly, yet sufficient incorporation to sequence and map replication sites.

We agree that the balance between possible replication stress and sufficient incorporation to profile DNA synthesis has to be achieved for Pu-seq experiments. We had performed the experiments in conditions which did not visually perturb cell growth. However, in the response to the reviewer's suggestion, we carefully monitored the DNA damage checkpoint, DNA synthesis and cell growth during the 0 to 48 hr of degradation of RNASEH2A (new

Supplementary Fig. 1). These additional data confirm that cell population growth and the level of DNA synthesis are not affected and demonstrate that the replication checkpoint, as monitored by Chk1 phosphorylation status, is not activated at the time of our sampling (48 hr).

2. What are the control lines used for Pu-seq analysis shown? WT polymerases with or without IAA induction? I think this should be stated in the main text.

For each individual experiment we induced RNASEH2A degradation in both the respective polymerase mutant and a wild-type polymerase cell line. The data derived from wild-type polymerase cell line in this condition are used as the control. As suggested by the reviewer, we clarify this point in the main text (please see the first paragraph of the section ‘Mapping polymerase usage across the genome’).

3. What does the polymerase usage look like not normalized to control lines? In theory wouldn't isolation / alkaline treatment from control lines (no mutant and/or no IAA induction), result in little to no gDNA? What's the rationale behind normalizing to a control line and not only to total read count? Signal amplification?

The reviewer is correct, without RNASEH2A degradation we would recover very little DNA. In the control cells (RNASEH2 degradation, Pol+) we recover >50ng and in the polymerase mutants we recover ~10 fold this amount. Thus, the amounts of DNA from both cell lines were sufficient for Illumina sequencing. Regardless of original cell lines, we use the same amount of DNA after the conversion of ssDNA to dsDNA for library preparation. Non-normalised profiles of polymerase mutants look very similar to those normalised with control cell lines at most of the genomic regions (we have added this data to Supplementary Fig. 2). However, our normalisation eliminates the influence of copy number variation in HCT116 cells. We also use the normalisation to minimise local variations which is not related to polymerase usage, e.g., biased output of sequence reads due to sequence content. By looking into the genome-wide data of representative regions closely, we found that local peaks (negative or positive) in non-normalised data (e.g. around 6465.5Mb and 65.5 Mb) disappear or less manifested after normalization.

4. Lines 225-230 (**Page 8 line 255~ in the revised manuscript**) / Extended Figure 6A (**main figure 4a in the revised version**): The authors conclude that ~20% of initiation zones don't map within regions ~30kb upstream of TSS of transcribed genes (“a-1” in Extended Figure 6). It could be interesting to see whether these 20% of genes are enriched for long genes, where persistent transcription machinery could be present. Comparing Main Figures 3C and 3D, it seems like the

association of initiation is somewhat stronger at TSS and TTS of very long genes vs very those of very highly transcribed genes. If these 20% of genes are enriched for longer genes, the conclusion that there is a subset of “transcriptionally active genes not necessarily associated with replication initiation” is maybe incomplete.

First, we clarify that, in this part, we described that 20% of active genes are not associated with an initiation zone at the upstream region. As pointed out by the reviewer, we compared the gene length of associated with initiation upstream TSS ('a-2' in Fig. 4a in the revised version) and those not associated ('a-1' in the same figure). As the histogram below shows, the average of gene length is larger in those genes with initiation, but their overall distribution is not dramatically different. In addition, we added the similar analysis to Fig. 4a to address the correlation between replication initiation around TSS/TTS and gene length (y-axis is the gene length instead of gene expression level, Fig. 4b). This result shows that initiation tends to associated with the large genes, but that a large gene does not necessarily associate with replication initiation around TSS or TTS. Thus, the length of gene is also not a critical determinant of replication initiation around genes.

5. Extended Figure 6B (**replaced with Fig 4c**): I think this figure is confusing. The authors describe 60% of initiation zones in both replicates don't map to either 30kb upstream of a TSS or downstream of a TTS. How was this analysis done? Do these 60% of origins represent less efficient origins? If they're trying to show the majority of origins are intergenic, that could be shown easily. 30kb is also a large region to be looking within and likely encompasses ~50% of genes. Are some of these origins intragenic? I think furthering this analysis would help the authors' argument that a significant portion of origins don't map to transcriptional regulatory units, or maybe it could be left out.

In this figure, we simply wanted to show the ratio of initiation zone which overlap with 30kb

upstream of a TSS or downstream of a TTS. As pointed out by reviewer, some upstream or downstream regions may include intragenic regions of neighbouring genes. Thus, to clarify this issue, we replaced this figure with one to show the kb-based ratio of area of initiation zones which overlap with transcribed regions and non-transcribed regions. We first obtained the length of initiation zones which overlap with 'transcribed gene regions' and 'non-transcribed regions' and then calculate their ratio to total length of initiation zones. The latter was further categorised into 'upstream/downstream region of actively transcribed regions' and 'others' (Fig. 4c).

6. Line 239 – 243 / Main Figure 4B (**Fig. 5b in the revised version**): It would be helpful for ease of understanding to mention the difference between fork index and the previously described initiation index, that the fork indices are mutually exclusive except for regions of overlap which correspond to the initiation index.

We modified the explanation of fork index (Page 9 line 275~) and highlighted the formula which represent the relationship between initiation index and fork index (Fig. 2d).

'As shown in the formula used to derive the initiation index (Fig. 2d), these profiles can be interpreted as separate 'initiation indices' for rightward and leftward moving forks that originate from non-overlapped polymerase profiles and thus independent parameters.'

7. Main figure 4D,E (**Fig. 5d and Supplementary Fig. 8b in the revised version**): To support the authors argument on the interpretation of the codirectional fork profile they should definitely show a subset of codirectional forks where there are no termination events immediately upstream of the TSS, if they argue the profile they see is a mixed signal of termination and initiation of multiple forks. It appears CD forks in general are associated with more termination than CV forks regardless of origin status at TSS (Extended Figure 6C [**Supplementary Fig. 8d in the revised version**], comparing "a-1" bottom and top). However, there still appears to be a relative termination within CD origins at TSSs (Extended Figure 6C, top, "a-2"). As it stands the data from extended figure 6C and figure 4D suggests that every single codirectional fork is associated with an upstream termination, which is a different mechanism than the authors propose in the text.

We agree that the termination of CD forks occurs in the vicinity of TSS sites, regardless of status of initiation. However, this does not necessarily mean termination occurs at all these regions. To address this issue, we added a heatmap which shows trends of fork index in the individual regions upstream of TSS (Fig. 5e, Supplementary Fig. 8c). This data clearly shows a heterogenous pattern in the fork index – in some regions, signals of initiation are exhibited and in other regions, termination occurs. This data supports 'this mixed signal of fork termination and initiation' around TSS.

8. Main Figure 5 (**Fig. 6 in the revised version**): One important control would be to show that this is uncoupling and not due to differing efficiency in mutant polymerase activity possibly at specific regions of the genome, or alternate polymerase usage.

We understand the reviewer's concern but we believe it is unlikely that the polymerase-domain mutated polymerase becomes inactive or inefficient at specific regions in the genome in a manner that is distinct from wild-type polymerase. If the mutation of the polymerase domain influences replication kinetics, the rate of DNA synthesis should become low anywhere in the genome; this is not the case in our system (Supplementary Fig. 1c). We also think that this issue is hard to examine experimentally.

Also, it is super surprising that uncoupling can last over 1 Mb in length. If this truly was the case, that is a lot of exposed ssDNA and checkpoint activation and single/double strand DNA breaks occurs within this region. Activation checkpoints would alter replication dynamics indirectly so it is difficult to know what is the "cause" and "effect" of these changes.

We would note that the effect on '1 Mb' are unlikely to result from a single continuous fork event. We rather think that the sum of effects in this region of a large number of forks are manifested as the uncoupling profile, since our experiments use a bulk of cells. In addition, as we have demonstrated in Supplemental Fig 1e that the checkpoint pathway is not induced in our condition, we believe that checkpoint activation is not the cause of replication perturbations we observe.

9. Line 349 (line 397~ in the revised version) / Figure 6B (**Fig. 7b in the revised version**): It appears that the low CI does correlate with a rightward fork initiation and leftward fork termination, contrary to the author's argument.

We apologise that our explanation was not entirely accurate. We modified this part and addressed that the profiles of fork dynamics do not account for the intensive change in coupling index because the local pattern of fork index does not match that of coupling index (the last paragraph of the result section).

'The local pattern low coupling index loci did not fully match with those of replication initiation or termination zones and thus fork dynamics does not account for low coupling index in these regions'

10. A graphical figure incorporating a model of transcription-replication collisions at CD and CV

forks, plus the distinct decoupling profiles at each of these TSSs could be helpful.

We add the graphics to depict the model in Supplementary fig 12.

Minor points:

1. Main Figure 1B: Add clarifying text above respective blots indicated AID vs AID2 system

We add the text as suggested.

2. Line 109-110 (**line 111~ in the revised version**): I'm not sure this last sentence is necessary, since there is no data in this study on mutant POLD overexpression. Also, the point isn't clear : ("mutant POLD does not provide essential function"); through what assays was this conclusion made?

We consider that many readers would also be interested in the situation of experimental set-up for Pol δ thus we wish to keep the description about the POLD1 mutant. But, as suggested by reviewer, we deleted this last sentence of the paragraph.

3. "Co-directional" is misspelled throughout the figures

Corrected.

4. It could be helpful to depict a termination event calculation similar to in figure 2B for ease of understanding how negative initiation values / fork indices correspond to termination events

As suggested by the reviewer, we add figures to depict the calculation of termination event (Fig. 2bd).

5. Line 262 (**line 307~ in the revised version**): "The combined pattern of CV and CD forks is consistent with the trend for fork initiation in both orientations 20kb upstream of TSS...". It would be useful to plot the 0% CV and CD lines at TSS and TTS seen in 4D (**Fig. 5d in the revised version**) on a separate graph in the supplement to more easily understand the switch from left/right fork indexes in Main Figure 4B (**Fig. 5b in the revised version**) to convergent / codirectional in Figure 4D and onward. This will allow readers to see the leftward index corresponds to convergent forks and right fork index corresponds to codirectional forks, and understand the mapped replication fork directionality without influence of transcription. This figure could be referenced after this clause on line 262-263.

As suggested by the reviewer, we added the separate graph of 0% CD and CV fork indices, which are referred as rightward and leftward fork respectively in Supplementary Fig. 8e.

6. Figure 4E (**Supplementary Fig. 8b in the revised version**): 0.75 label is negative
Corrected.

7. Extended Figure 8A (**Supplementary Fig. 9a in the revised version**): what do 0-50% and 50-100% refer to? Percentile of origin efficiency?

This is percentile of initiation index. We add the explanation in the legend (Supplementary Fig. 9 in the revised manuscript).

8. Main Figure 5D (**Fig. 6d in the revised version**): label scale bar missing

We add the label 'The number of genomic location (unit: 1kb)' (Fig. 6d in the revised manuscript)

9. Line 329 – 330: sentence typo

Thank you for point this out. Corrected.

10. Figure 6B (**Fig. 7b in the revised version**): grey boxes interrupting view of zoomed in
These grey bars mask regions where uniquely mapped reads are very few (less than 5 in both watson and crick strands) and thus we were unable to calculate coupling index. We clarify this point in the figured legend.

11. Line 432 – 433: missing citation ([https://www.cell.com/cell-reports/pdf/S2211-1247\(21\)00072-3.pdf](https://www.cell.com/cell-reports/pdf/S2211-1247(21)00072-3.pdf))

We appreciate the reviewer to guide us to the recent publication. We delete this sentence and cited this study in discussion (line 444 citation 41).

REVIEWERS' COMMENTS

Reviewer #1 (Remarks to the Author):

The authors have adequately addressed my concerns. Repli-seq data from HCT116 cells should be available from several labs at this point, however this is a minor concern.

Reviewer #2 (Remarks to the Author):

The authors have addressed most of the comments satisfactorily. Just one note, it's good to see that the initiation zones defined by Pu-seq have some overlap with the ones described by OK-seq. However, it seems that the number of initiation zones determined by different techniques varies greatly. May the authors please provide possible reasons for that?

Reviewer #3 (Remarks to the Author):

The revised manuscript by Koyanagi et al has sufficiently addressed most of the reviewers' concerns.

We thank all referees for positive responses to our revised manuscript. Below we described our responses to reviewers (in bold text). Changes to the manuscript were shown using the tracking function in MS word.

REVIEWERS' COMMENTS

Reviewer #1 (Remarks to the Author):

The authors have adequately addressed my concerns. Repli-seq data from HCT116 cells should be available from several labs at this point, however this is a minor concern.

As reviewer mentioned, multiple repli-seq data in HCT116 cells were published. However, we found that only high-resolution repli-seq from Zao et al (2020) has comparable resolution to Pu-seq data.

Reviewer #2 (Remarks to the Author):

The authors have addressed most of the comments satisfactorily.

Just one note, it's good to see that the initiation zones defined by Pu-seq have some overlap with the ones described by OK-seq. However, it seems that the number of initiation zones determined by different techniques varies greatly. May the authors please provide possible reasons for that?

In term of difference between Pu-seq and OK-seq, it may depend on the cell line used. Consistent with this thought, patterns of initiation zone are distinct even between two OK-seq data in HeLa cells and GM06990 cells, suggesting the genome-wide distribution of initiation zone varies depending on cell lines.

Compared to Pu-seq and OK-seq, other techniques (SNS-seq, bubble-seq and ini-seq) predicted the greater number of initiation sites. We thus presume that OK-seq and Pu-seq, which detect for reciprocal pattern of leading/lagging strand DNA synthesis, are relatively conservative methods to identify replication initiation sites.

However, these discussions largely rely on our speculation and thus we will avoid to describe them in the context.

Reviewer #3 (Remarks to the Author):

The revised manuscript by Koyanagi et al has sufficiently addressed most of the reviewers' concerns.

Thank you for all valuable suggestions.